# FCMI-YOLO: An efficient deep learning-based algorithm for real-time fire detection on edge devices

**Junjie Lu**[1], **Yuchen Zheng**[1], **Liwei Guan**[2]*, **Bing Lin**[2], **Wenzao Shi**[1], **Junyan Zhang**[1], **Yunping Wu**[1]

**1** College of Photonic and Electronic Engineering, Fujian Normal University, Fujian, China, **2** College of Physics and Energy, Fujian Normal University, Fujian, China

* guanlw@fjnu.edu.cn

**Data availability statement:** The data used in this study are publicly available on GitHub at the

## Abstract

The rapid development of Internet of Things (IoT) technology and deep learning has propelled the deployment of vision-based fire detection algorithms on edge devices, significantly exacerbating the trade-off between accuracy and inference speed under hardware resource constraints. To address this issue, this paper proposes FCMI-YOLO, a real-time fire detection algorithm optimized for edge devices. Firstly, the FasterNext module is proposed to reduce computational cost and enhance detection precision through lightweight design. Secondly, the Cross-Scale Feature Fusion Module (CCFM) and the Mixed Local Channel Attention (MLCA) mechanism are incorporated into the neck network to improve detection performance for small fire targets and reduce resource consumption. Finally, the Inner-DIoU loss function is proposed to optimize bounding box regression. Experimental results on a custom fire dataset demonstrate that FCMI-YOLO increases mAP@50 by 1.5%, reduces parameters by 40%, and lowers GFLOPs to 28.9% of YOLOv5s, demonstrating its practical value for real-time fire detection in edge scenarios with limited computational resources. The core code and dataset are available at https://github.com/ JunJieLu20230823/code.git.

## 1 Introduction

Fire is a frequent disaster that poses a significant threat to public safety and social development. Its progression is typically divided into four stages: ignition, growth, full development, and decay [1]. Each stage has distinct characteristics that require corresponding prevention and specific control measures to effectively manage the fire. The most effective strategy for fire prevention and suppression is to detect and extinguish fires during the incipient stage, preventing the fire from escalating into rapid growth or full development stages.

Fire detection systems have transitioned from traditional manual observation and routine patrols to IoT-based technologies [2]. By deploying temperature sensors [3], smoke sensors [4], and light sensors [5], these systems enable real-time monitoring of environmental changes, such as temperature, smoke particles, and spectral shifts, to achieve automated and intelligent fire detection and alerts. However, the limited coverage of sensor deployments

following URL: https://github.com/JunJieLu20230823/code. The repository contains the full dataset used for model training, validation, and evaluation, encompassing a wide range of image samples reflecting various styles, scenes, and complex visual features relevant to fire detection. All relevant data are freely accessible without restriction, ensuring transparency and reproducibility of the research.

**Funding:** This study is supported by the following grants: Quanzhou Science and Technology Program (No. 2024NS001) Principal Investigator: Liwei Guan The funder had a role in preparation of the manuscript. General Program of Education and Teaching Research for Undergraduate Universities of Fujian Province (No. FBJG20220058) Principal Investigator: Yunping Wu The funder had a role in preparation of the manuscript. Natural Science Foundation of Fujian Province (No. 2021J01162) Principal Investigator: Wenzao Shi The funder had a role in preparation of the manuscript.

**Competing interests:** The authors have declared that no competing interests exist.

leads to monitoring blind spots, and the poor sensor density directly impacts the success of fire detection in the ignition stage.

With the development of advanced object detection approaches, the vision-based fire detection approaches have gained significant attention [6,7]. These approaches utilize cameras or other image acquisition devices to capture real-time video streams and detect fires through image processing and analysis algorithms [8–10]. Compared to IoT sensor-based technologies, vision-based approaches offer significant advantages in spatial coverage, sensitivity, and cost efficiency, particularly in large-scale monitoring and dynamic environments [7].

Deep learning-based object detection approaches have revolutionized fire detection as a novel advancement in computer vision and supervised learning. These approaches leverage neural network models to automatically learn and extract multi-layered abstract high-dimensional features from extensive datasets, offering superior robustness and adaptability. This enables them to effectively handle complex environmental interferences while achieving exceptional detection accuracy and generalization capabilities. However, as the accuracy of deep learning-based object detection approaches continues to improve, the computational complexity and resource demands of object detection models have increased significantly, severely constraining their feasibility for deployment on low-power embedded devices. This technical bottleneck has become even more pronounced in the IoT era, where the large-scale proliferation of embedded devices necessitates more efficient solutions. Cao et al. [11] proposed the YOLO-SF algorithm, which combines instance segmentation technology with YOLOv7 while incorporating the MobileViTv2 module and the Convolutional Block Attention Module (CBAM) to enhance fire feature extraction capabilities. Although the accuracy of detection increased by 4%, the number of parameters doubled, and the Frames Per Second (FPS) nearly halved, which poses a significant obstacle to deployment on mobile embedded devices. To enhance multi-scale fire feature capture, Wang et al. [12] optimized the feature pyramid network in YOLOv8 using an FSPPF structure, introduced an additional small-object detection layer to extend multi-scale perception, and incorporated Dynamic Snake Convolution (DSC) to enhance feature fusion. While this algorithm improves fire detection mAP@0.5 by 1.9%, it also increases model parameters by 1.6 times and FLOPs by 2.1 times, imposing a significant burden on real-time processing in edge computing environments. He et al. [13] removed the FPN-PAN structure from the YOLOv5 neck network and merged the three detection heads into a single-head prediction head, significantly reducing model parameters and achieving a 29ms per-frame inference speed on edge devices. However, this algorithm leads to a 3.6 percentage point decrease in Average Precision (AP), making it less robust in complex environments.

Although existing deep learning methods have been widely applied to fire detection, they often face a trade-off between accuracy and inference speed. The proposed FCMI-YOLO, an enhancement of YOLOv5s, not only improves the accuracy of fire detection but also optimizes inference speed on edge devices. In comparison to the studies mentioned above, our method strikes a superior balance between accuracy and inference speed, featuring a lightweight design, efficient deployment, and real-time detection capabilities. The contributions are as follows:

- A lightweight FasterNext module was designed to replace the C3 module in the backbone network, which reduces both the number of parameters and computational load while enhancing the model's feature extraction capability in complex environments.
- The neck network was optimized by integrating the CCFM and the MLCA mechanism, which first employed lightweight convolutions to extract features from the deep network

and then utilized the MLCA mechanism to focus on key information, effectively enhancing the model's detection performance for small targets.

- The Inner-IoU loss function was introduced to optimize bounding box regression. By incorporating auxiliary bounding boxes, the model's sensitivity to fire scale variations was improved, resulting in enhanced localization accuracy.
- A dedicated fire dataset was constructed to support the algorithm's training and evaluation. Experimental results demonstrated that FCMI-YOLO outperforms other YOLO-based algorithms, achieved superior comprehensive performance, and enabled real-time monitoring of medium- and long-range fires on edge devices.
- FCMI-YOLO was deployed on the Orange Pi 5 Plus edge device, utilized an asynchronous multi-threading strategy to accelerate inference speed, and achieve efficient real-time detection of medium- and long-range fire sources.

The remainder of this paper is organized as follows: Sect Sect 2 provides a comprehensive review of existing deep learning-based methods for fire detection. Sect Sect 3 presents a detailed description of the improvements made to the YOLOv5s algorithm and introduces the FCMI-YOLO algorithm, specifically designed for enhanced fire detection. Sect Sect 4 evaluates the performance metrics of the FCMI-YOLO algorithm on both personal computers (PC) and edge devices. Finally, Sect Sect 5 summarizes the research contributions and concludes the paper.

## 2 Related work

Vison-based fire detection methods can be broadly divided into two categories: methods based on handcrafted feature extraction and deep learning. Handcrafted feature extraction methods detect fire by extracting static features such as color, texture, and blur level, as well as dynamic features like shape changes, flickering, and motion direction. These methods rely on classical computer vision methods, including probability density functions, color space analysis (e.g., YUV [14] and HSV [15]), and texture analysis (e.g., SIFT [16] and HOG [17]), combined with classifiers such as SVM and AdaBoost for fire detection. However, they are susceptible to environmental conditions, such as variations in lighting and airflow, leading to high false positives and limited robustness in long-range detection or complex scenarios. Deep learning methods have addressed the limitations through end-to-end feature learning mechanisms. Convolutional neural network (CNN)-based methods can automatically extract multi-scale abstract features from large-scale data and construct hierarchical representations that are robust to environmental interference [18,19]. As a result, they have been widely applied to domains including autonomous driving [20], smart agriculture [21], security surveillance [22], and industrial IoT [23–26].

In the context of fire detection, CNN-based object detection algorithms can be divided into two categories: two-stage and one-stage algorithms [8]. Two-stage detection algorithms, represented by Faster-RCNN [27] and Fast-RCNN [28], generate candidate regions and progressively optimize target recognition results, achieving high detection accuracy. However, these algorithms involve extensive redundant computations during region generation and classification, resulting in slower inference speeds that fail to meet real-time detection requirements. In contrast, one-stage detection algorithms, such as YOLO [29] and SSD [30], simplify the detection pipeline by performing target classification and bounding box regression directly on the image, bypassing redundant steps like region proposal generation and classification optimization. These algorithms significantly reduce computational overhead and shorten information processing pathways, providing a notable advantage in processing speed.

Among these, YOLO stands out for its superior inference speed and detection accuracy, making it the preferred choice for deployment on edge devices, particularly in fire detection applications with high demands for real-time performance and accuracy [31–34].

Recent YOLO-based studies have focused on improving both accuracy and efficiency in fire detection tasks. Jiang et al. [35] proposed the DG-YOLO model based on YOLOv8, which integrates a deformable attention mechanism, a lightweight feature extraction module (GSC2f) for cross-scale edge information fusion, and a dedicated small-target detector to enhance texture detail capture. Under complex background interference, the model achieves a 10.7% improvement in mAP@0.5. To address the limitations of single model feature representation, Xu et al. [36] proposed a fire detection method that integrates YOLOv5 and EfficientDet, with EfficientDet capturing global features to reduce false positives caused by overemphasis on local details, resulting in a 2.5%-10.9% improvement in detection performance and a 51.3% reduction in false positives. For complex and dynamic scenarios, Luan et al. [37] proposed a lightweight fire detection algorithm based on YOLOX, which integrates a multi-level feature extraction structure (CSP-ML) and the CBAM attention mechanism to enhance small-target recognition. By leveraging multi-branch feature fusion, the model captures both the positional sensitivity of shallow high-resolution features and the semantic abstraction of deep features, resulting in a 6.4% improvement in mAP@0.5 under smoke occlusion and dynamic background interference. Zhao et al. [38] proposed a Fire Segmentation-Detection Framework (FSDF), which enhances fire representation by jointly extracting image color and texture features, and integrates YOLOv8 with Vector Quantized Variational Autoencoders (VQ-VAE) to enable supervised target localization and unsupervised fire feature learning, respectively, thereby improving recognition accuracy in complex scenarios.

To enable efficient deployment on edge devices, Wang et al. [39] proposed Light-YOLOv4, which replaces the original backbone network with MobileNetv3 to reduce parameters, incorporates a depthwise separable attention module that combines depthwise separable convolutions and a coordinate attention mechanism to minimize computational redundancy and introduces a Bidirectional Feature Pyramid Network (BiFPN) to enhance multi-scale feature interaction, resulting in an 80.9% reduction in parameters while maintaining 85.64% accuracy on resource-constrained hardware devices. To further enhance lightweight fire detection, Huang et al. [40] proposed YOLO-ULNet, which employs Grouped Channel Shuffle (GCS) units within Light-weight Feature Extraction (LFE) units to reduce parameters, incorporates a Multipath Aggregation Feature Pyramid (MAFP) structure to enable efficient multi-dimensional feature fusion, and integrates channel pruning and feature distillation techniques to compress the model, resulting in a detection speed of 24.57 FPS and 74.50% accuracy on a Raspberry Pi 4B, with only 0.19M parameters and 0.4 GFLOPs.

The studies above provide valuable insights into the development of fire detection models but also reveal certain limitations. Models that emphasize feature enhancement often introduce additional complexity, leading to increased parameter counts and computational costs that hinder real-time deployment. In contrast, lightweight models tend to suffer from reduced detection precision, especially when dealing with small or distant targets. Striking a balance between detection accuracy and inference speed remains a core challenge, particularly under the resource constraints of edge devices. To address these challenges, this paper proposes FCMI-YOLO, which achieves a trade-off between accuracy and inference speed on edge devices, effectively mitigating the issues of high resource consumption and low detection precision.

## 3 Proposed methods and model architecture

This section provides an overview of the FCMI-YOLO algorithm, highlighting its key improvements and optimizations for real-time fire detection on edge devices. Based on YOLOv5s architecture, FCMI-YOLO aims to enhance detection accuracy while minimizing computational complexity, making it suitable for resource-constrained environments.

Sect introduces the overall architecture, describing how the input layer, backbone, neck, and detection head work together for efficient fire detection. Sect , the FasterNext module is presented, replacing the C3 module in the backbone to reduce complexity while maintaining feature extraction performance. Sect details integrating the Cross-Scale Feature Fusion Module (CCFM) and the Mixed Local Channel Attention (MLCA) mechanism into the neck network to improve detection accuracy, particularly for small fire targets, while minimizing resource usage. In Sect , the Inner-IoU loss function is also discussed, optimizing bounding box regression to enhance the model's sensitivity to variations in fire target size and shape. These improvements collectively form a robust, efficient, and real-time fire detection solution for edge devices.

### 3.1 FCMI-YOLO

The FCMI-YOLO algorithm is based on the YOLOv5s architecture, which consists of four main components: the input layer, backbone network, neck network, and detection head, working together to achieve end-to-end object detection [41]. The input layer enhances the model's generalization ability through data normalization and augmentation; the backbone network extracts multi-scale features using hierarchical down-sampling; the neck network fuses both detailed and semantic features from the backbone network; and the detection head uses an anchor-based mechanism to generate multi-scale feature maps for precise detection of objects of different sizes.

To enhance the performance of YOLOv5s on resource-constrained edge devices, while meeting the requirements for high accuracy and real-time fire detection, FCMI-YOLO introduces several key optimizations to the overall architecture. As shown in Fig 1, the improvements include the following key aspects. First, in the backbone network, all C3 modules are replaced with the newly designed FasterNext module. Based on the design philosophy of FasterNet, this module significantly reduces the model's complexity while maintaining strong feature extraction capabilities. Second, at the neck network, a Cross-Scale Feature Fusion Module (CCFM) is introduced, which strengthens the fusion of multi-level features through lightweight convolution operations, effectively optimizing the representation of multi-scale features. Furthermore, a Mixed Local Channel Attention (MLCA) mechanism is added, combining both spatial and channel attention, enabling the model to focus more accurately on key regions, thus improving detection performance for small fire targets. Finally, in terms of the loss function, the traditional CIoU loss is replaced with Inner-DIoU, which introduces an auxiliary bounding box mechanism to enhance the model's sensitivity to variations in fire target size and irregular shapes, significantly improving bounding box localization accuracy.

### 3.2 FasterNext network

The backbone network of YOLOv5s uses the C3 module, which consists of three CBS modules and several Bottleneck modules. Although the Bottleneck module helps mitigate the issue of gradient vanishing, it still results in some loss of feature information in complex scenarios. Additionally, its large number of parameters poses challenges for deployment on edge devices. To address these issues, the design principles of the FasterNet network [42] were

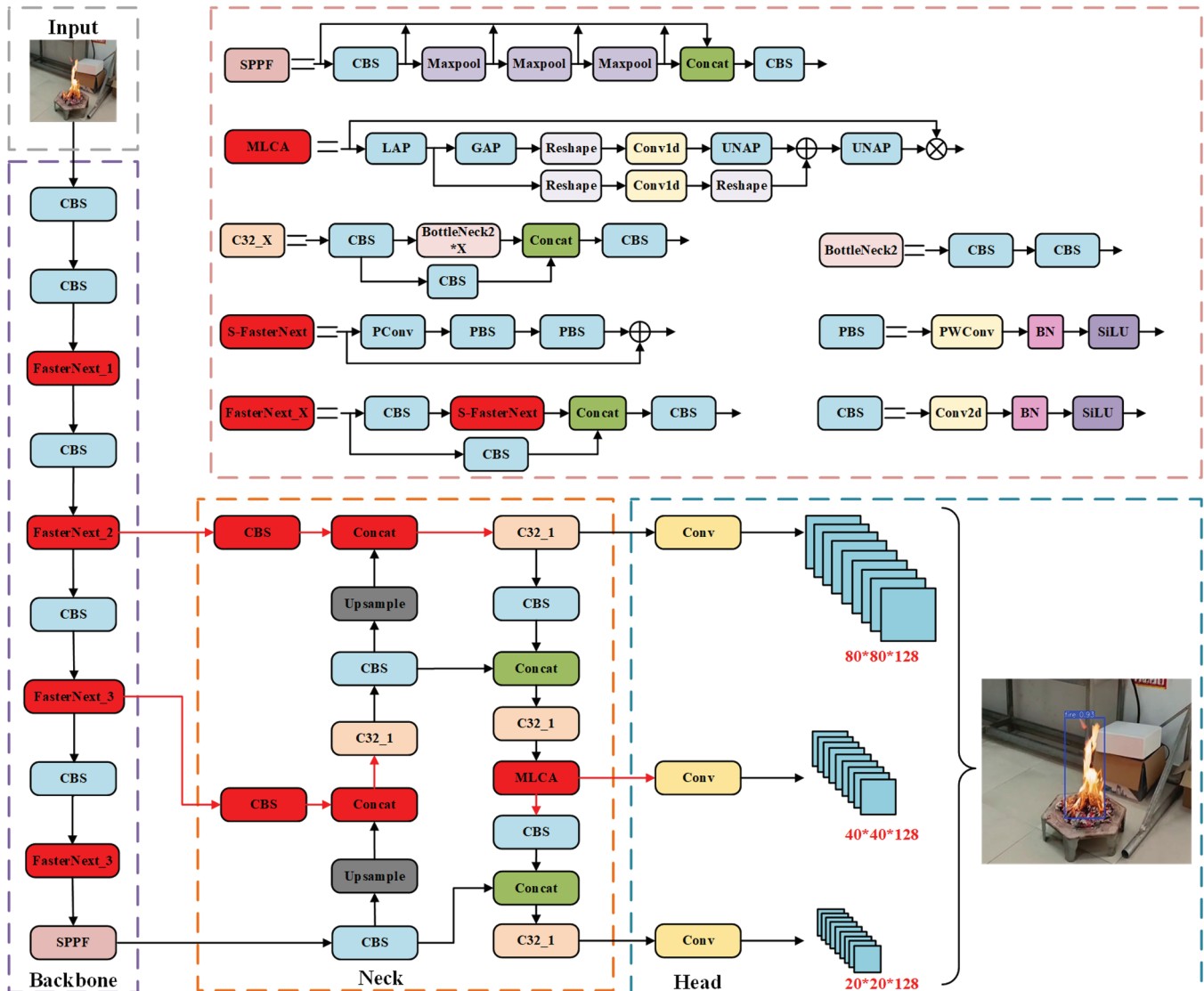

**Fig 1. Overall network architecture of FCMI-YOLO.**

adopted, leading to the development of a novel network structure named FasterNext, which replaces the C3 module in the backbone network. The structures of FasterNet, S-FasterNet, and FasterNext are shown in Fig 2(**a**), 2(**b**), and 2(**c**), with S-FasterNet being a key component of the FasterNext network.

As shown in Fig 2(**a**), the FasterNet network consists of a PConv layer and two PWConv layers, forming a reverse residual block. The working principle of the PConv layer is illustrated in Fig 3. Unlike conventional convolution, PConv only performs convolution operations on a subset of input channels, while the remaining channels remain unchanged, making more efficient use of computational resources. To achieve continuous or regular memory access, PConv uses either the first or last set of consecutive channels as representatives of the entire feature map during computation. Assuming the input and output feature maps have the same number of channels, the FLOPs (floating point operations) of PConv can be calculated

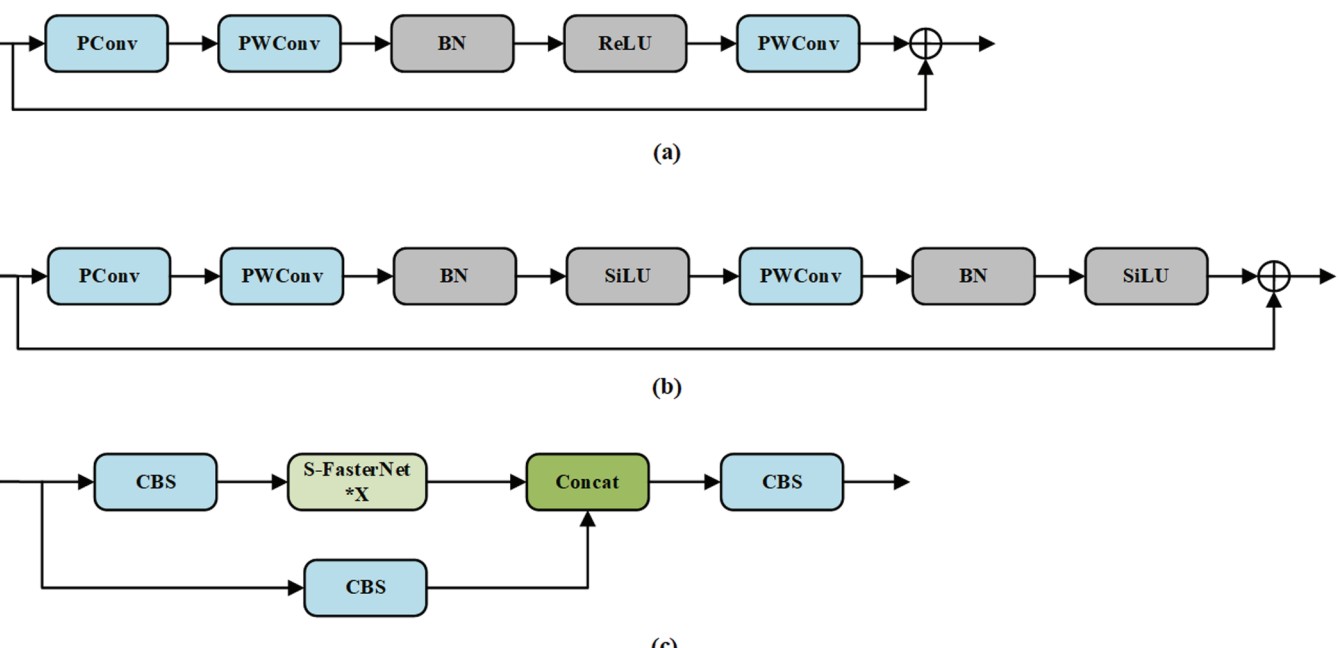

**Fig 2. The structure of FasterNext.** (a) FasterNet. (b) S-FasterNet. (c) FasterNext.

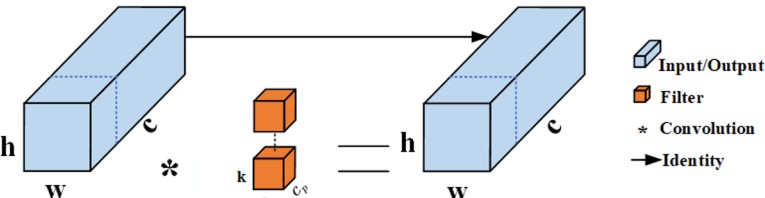

**Fig 3. The principle of Partial Convolution.**

as: $h \times w \times k^2 \times C_p^2$, where $h$ and $w$ represent the height and width of the feature map, $k$ is the kernel size and is the number of channels involved in the partial convolution. When PConv uses only one-quarter of the channels, its FLOPs are reduced to 1/16 of a regular convolution. To ensure feature diversity and low latency, batch normalization, and ReLU activation layers are placed exclusively between the two PWConv layers. However, this design might be insufficient for handling complex feature extraction tasks.

According to Fig 2(**b**), the S-FasterNet module replaces the original ReLU activation function with SiLU and applies batch normalization and SiLU after each PWConv layer to enhance nonlinear representation and feature extraction. This design enhances the network's nonlinear transformation capability and improves feature extraction performance under complex scenarios. ReLU is one of the most common activation functions in deep neural networks, favored for its computational simplicity and sparse activation properties. It is defined in Eq 1.

$$ReLU(x) = \max(0, x) \tag{1}$$

However, ReLU has several limitations. When the input is negative, the gradient becomes zero, which may lead to the permanent deactivation of certain neurons during training. Moreover, its non-differentiability at the origin and complete suppression of negative values can hinder gradient propagation and reduce the network's expressive capacity. To overcome these issues, the SiLU activation function is employed, defined in Eq 2.

$$SiLU(x) = \frac{x}{1 + e^{-x}} \tag{2}$$

As shown in Fig 4, SiLU offers a smooth and continuous derivative, enabling a more stable gradient flow. Unlike ReLU, it provides a moderated response to negative inputs rather than discarding them entirely, thereby improving feature retention and model expressiveness. In fire detection tasks, the nonlinear behavior of SiLU allows more accurate modeling of fire characteristics, leading to better detection performance in complex environments.

According to Fig 2(**c**), the FasterNext network mainly consists of two parallel branches. The first branch contains one CBS module and multiple stacked S-FasterNet modules, while the second branch passes through a single CBS module. The two branches are merged using a Concat operation, and finally, the output feature map is generated through a CBS module. As shown in Table 1, FasterNext significantly reduces the number of parameters compared to the C3 module, achieving approximately a 31% reduction while maintaining the same input image size and number of channels.

## 3.3 Improvements in the neck network

### 3.3.1 Cross-scale feature fusion module.
In the neck network of YOLOv5s, the Path Aggregation Network (PAN) structure is employed to fuse target features at different scales,

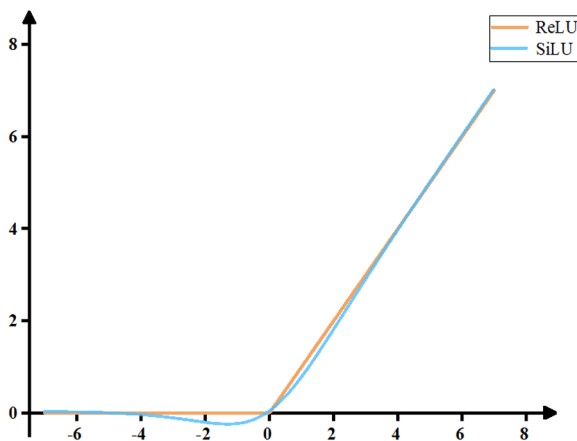

**Fig 4. Comparison curve of ReLU and SiLU activation functions.**

**Table 1. Parameters of the FasterNext and C3.**

| Model Name | Img Size | Channels | Parameters |
|---|---|---|---|
| C3 | 640 × 640 | (256,256) | 296448 |
| FasterNext | 640 × 640 | (256,256) | 202240 |

thereby enhancing detection performance. However, this feature fusion approach has limitations, including high computational complexity, excessive memory consumption, and insufficient capability to capture small target information. To address these issues, a Cross-Scale Feature Fusion Module (CCFM) [43] was introduced.

CCFM constructs multi-scale feature interaction paths using lightweight convolution operations, achieving cross-layer feature reuse while reducing computational complexity. This design effectively preserves high-resolution spatial details, mitigating the loss of small-object information caused by feature map downsampling. Additionally, it employs a strategy based on learned weights to automatically adjust the feature fusion approach. This grants the model a high degree of flexibility, allowing it to adapt the integration of features from different scales according to the demands of fire detection tasks. By enabling collaborative perception of local details and global semantics, this design enhances the model's capability to distinguish complex fire characteristics while ensuring detection accuracy. This design ensures detection accuracy while adopting a lightweight architecture suitable for resource-constrained edge devices, providing an efficient solution for real-time fire detection tasks.

**3.3.2 Mixed local channel attention mechanism.** The attention mechanism is designed to enable models to focus more precisely on critical information, thereby enhancing performance and efficiency. However, traditional channel attention mechanisms, while effective in amplifying feature representation along the channel dimension, exhibit limitations in capturing spatial information, which can compromise detection accuracy. In contrast, spatial attention mechanisms can capture local details of images but often fail to focus accurately on critical regions due to overly uniform attention distribution. Additionally, their high computational complexity and large parameter count restrict their application in edge devices.

As illustrated in Fig 5, the Mixed Local Channel Attention (MLCA) mechanism [44] addresses these issues by dividing input feature maps into multiple local regions, combining spatial and channel information processing, and utilizing $1 \times 1$ convolutions to significantly reduce computational overhead and parameter count. Compared to other attention mechanisms, MLCA incorporates both spatial and channel information, optimizing resource utilization while improving model robustness. Consequently, MLCA is introduced after the CCFM to further enhance the model's performance. The synergy between feature fusion and the attention mechanism substantially improves the model's ability to learn fire-specific features and detect small fire targets, providing robust technical support for mid-to long-range fire detection.

The structure of the MLCA mechanism is shown in Fig 6. First, the input feature map undergoes Local Average Pooling (LAP), which reduces its dimensions to $1 \times 128 \times 5 \times 5$. Then, the processed feature map is split into two paths. The first path extracts global features through Global Average Pooling (GAP), while the second path reshapes the feature map to extract local features. Next, the features from both paths undergo 1D convolution processing, and the feature maps are then restored to $1 \times 128 \times 5 \times 5$ dimensions using Unpooling and reshaping operations. Finally, the features from both paths are fused, and the resulting feature map is restored to the original dimensions of the input feature map via a Unpooling operation to generate the final output.

## 3.4 Improved loss function

The YOLOv5s algorithm originally uses the CIoU loss function to evaluate the quality of predicted bounding boxes. This function enhances localization accuracy by evaluating the

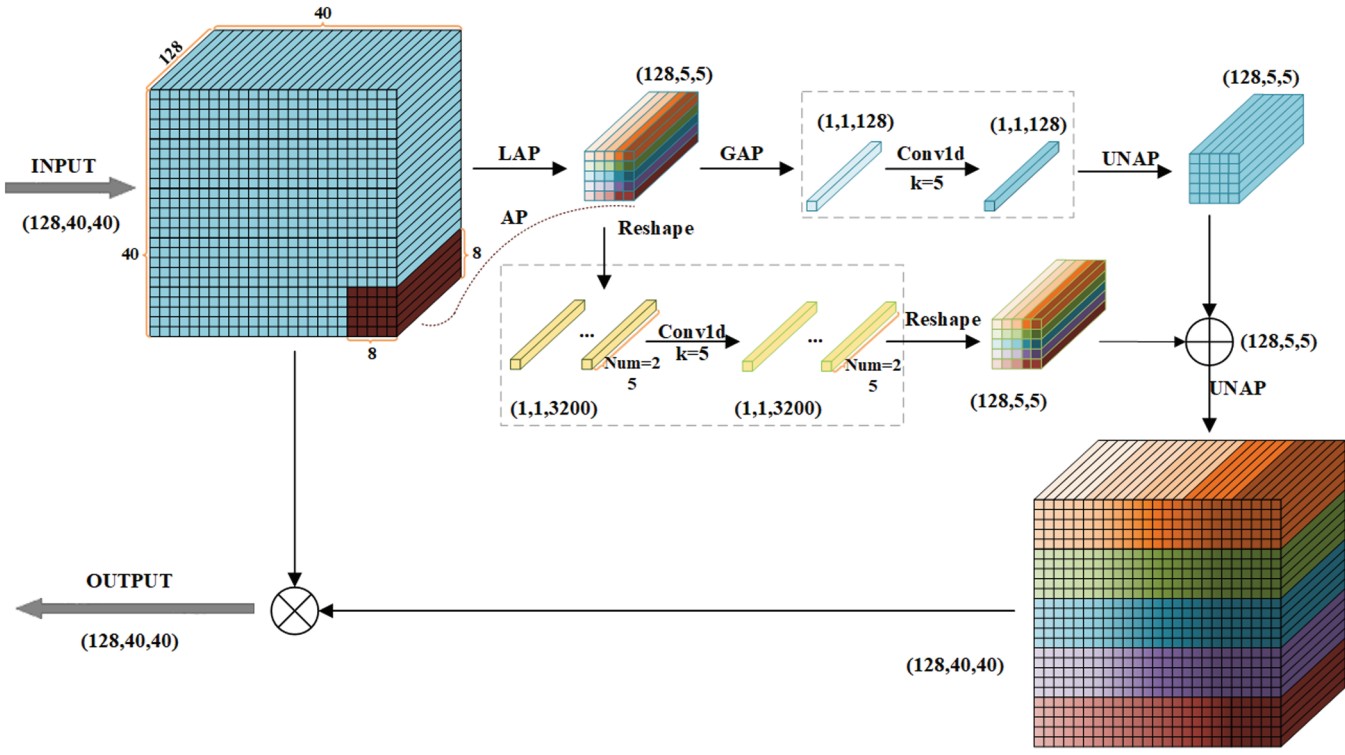

**Fig 5. The principle of MLCA mechanism.**

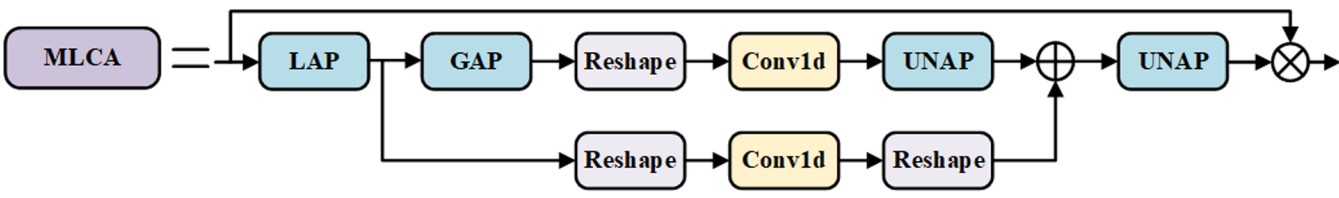

**Fig 6. The structure of MLCA mechanism.**

overlap area between the predicted box and the ground truth, the distance between their center points, and the consistency of their aspect ratios. The definition of CIoU is shown in Eqs (3)–(6).

$$L_{CIoU} = 1 - IoU + \frac{\rho\left(b, b^{gt}\right)}{c^2} + \alpha\nu \tag{3}$$

$$IoU = \frac{inter}{union} \tag{4}$$

$$\alpha = \frac{\nu}{(1 - IoU) + \nu} \tag{5}$$

$$\nu = \frac{4}{\pi^2} \left( \arctan \frac{w^{gt}}{h^{gt}} - \arctan \frac{w}{h} \right)^2 \tag{6}$$

Where $IoU$ represents the ratio between the intersection and the union of the predicted bounding box and the ground truth box, measuring the degree of spatial overlap. The $\rho(b, b^{gt})$ denotes the euclidean distance between the center points of the predicted box and the GT box, used to penalize positional deviation. The $\alpha$ is a dynamic weighting factor that adaptively adjusts the contribution of the aspect ratio term based on the $IoU$ and shape difference. The $\nu$ quantifies the discrepancy in aspect ratio between the predicted and GT boxes by calculating the difference in the arctangent of their respective width-to-height ratios. Specifically, when the $IoU$ is low, $\alpha$ becomes smaller to focus more on improving the overlap region, and when the IoU is high, $\alpha$ increases to emphasize shape refinement. By jointly optimizing the overlap area, center distance, and aspect ratio, the CIoU loss function achieves improved localization accuracy and overall object detection performance.

However, the CIoU loss function shows limitations in fire detection, as its reliance on aspect ratio consistency makes it less effective at handling small or irregularly shaped fire targets, leading to insufficient sensitivity to scale variations, which in turn results in detection errors and slows down model convergence.To address these issues, the width-height ratio adjustment factor in CIoU is removed, which leads to the adoption of the DIoU loss function, further optimized by incorporating the Inner-IoU [45], resulting in the formulation of the Inner-DIoU loss function. The Inner-DIoU is defined in Eqs (7)–(9).

$$L_{DIoU} = 1 - IoU + \frac{\rho(b, b^{gt})}{c^2} \tag{7}$$

$$L_{Inner-IoU} = 1 - IoU^{inner} \tag{8}$$

$$L_{Inner-DIoU} = L_{DIoU} + IoU - IoU^{inner} \tag{9}$$

The Inner-IoU loss function optimizes detection performance by introducing a scale-scaling factor, $ratio$, which adjusts the aspect ratio of the auxiliary bounding box. As illustrated in Fig 7, when the $ratio$ equals 1, the auxiliary bounding box is equal to the actual bounding box. If the $ratio$ is less than 1, the auxiliary bounding box is smaller than the actual bounding box, facilitating faster convergence for high $IoU$ samples. Conversely, when the $ratio$ is greater than 1, the auxiliary bounding box is larger than the actual bounding box, which accelerates the regression of low $IoU$ samples. Eqs (10)–(13) define the boundary parameters of the Inner Box.

$$b_l^{gt} = x_c^{gt} - \frac{w^{gt} \times ratio}{2}, \quad b_r^{gt} = x_c^{gt} + \frac{w^{gt} \times ratio}{2} \tag{10}$$

$$b_t^{gt} = y_c^{gt} - \frac{h^{gt} \times ratio}{2}, \quad b_b^{gt} = y_c^{gt} + \frac{h^{gt} \times ratio}{2} \tag{11}$$

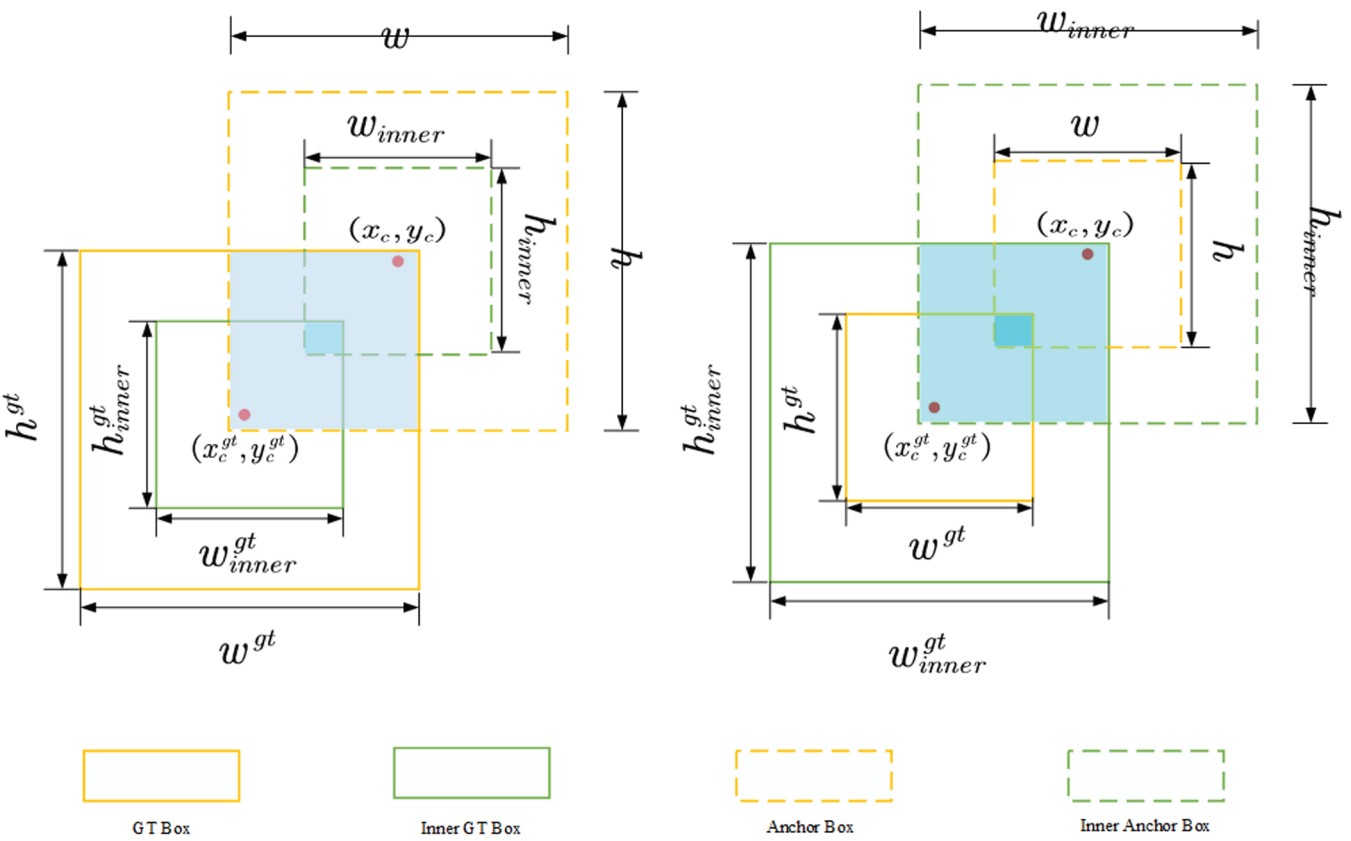

**Fig 7. Schematic diagram of Inner-IoU.**

$$b_l = x_c - \frac{w \times ratio}{2}, \quad b_r = x_c^{gt} + \frac{w \times ratio}{2} \tag{12}$$

$$b_t = y_c - \frac{h \times ratio}{2}, \quad b_b = y_c^{gt} + \frac{h \times ratio}{2} \tag{13}$$

Where $\left(x_c^{gt}, y_c^{gt}\right)$ represent the center coordinates of the GT box, and $(x_c, y_c)$ represent the center coordinates of the anchor box. $b_l^{gt}$, $b_r^{gt}$, $b_t^{gt}$, and $b_b^{gt}$ represent the left, right, top, and bottom boundary coordinates of the Inner GT Box, respectively. Similarly, $b_l$, $b_r$, $b_t$, and $b_b$ represent the left, right, top, and bottom boundary coordinates of the Inner Anchor Box, respectively.

In Eqs (14)–(16), the definition of Inner-IoU is provided.

$$inter^{inner} = \left(\min\left(b_r^{gt}, b_r\right) - \max\left(b_l^{gt}, b_l\right)\right) \times \left(\min\left(b_b^{gt}, b_b\right) - \max\left(b_t^{gt}, b_t\right)\right) \tag{14}$$

$$union^{inner} = \left(w^{gt} \times h^{gt}\right) \times (ratio)^2 + (w \times h) \times (ratio)^2 - inter \tag{15}$$

$$IoU^{inner} = \frac{inter^{inner}}{union^{inner}} \tag{16}$$

Where $inter^{inner}$ denotes the intersection of the Inner GT box and Inner anchor box, while $union^{inner}$ represents their union. The Inner-IoU is defined as the ratio of $inter^{inner}$ to $union^{inner}$.

## 4 Experiments and analysis

### 4.1 Dataset

The performance of fire image recognition algorithms is heavily dependent on the quality of the dataset. However, the field of fire detection faces significant challenges, including insufficient sample size, imbalanced sample distribution, and limited background diversity. Additionally, publicly available video and image datasets are scarce, and there is a lack of authoritative standard datasets [10].

To address this, the fire image dataset was collected through three approaches:

1. Selecting high-resolution images from publicly available datasets such as BoWFire [46], FASDD [47], and COCO2017 [48].
2. Collecting high-quality fire images from the internet using web scraping techniques.
3. Creating custom fire videos and extracting images from each frame.

After data collection, manual filtering and data augmentation techniques, including scaling and stitching, were applied to enhance data diversity. The dataset was then annotated using the LabelImg tool. Finally, the dataset was split into training, validation, and test sets with a ratio of 7:1.5:1.5. Fig 8 illustrates the data distribution, and Table 2 provides detailed statistical information on the images.

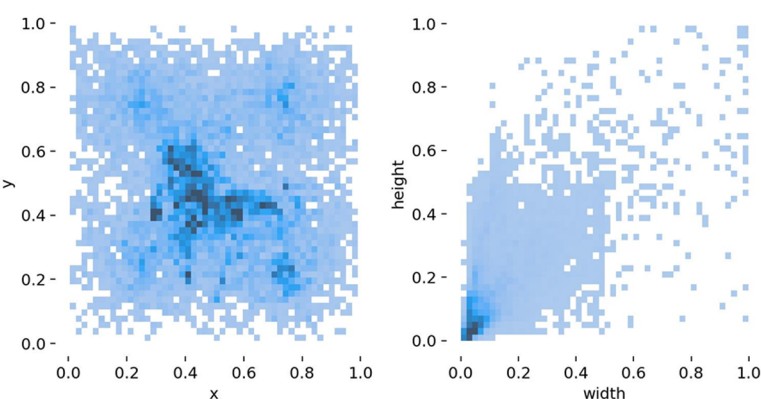

**Fig 8. Distribution of the dataset.**

Table 2. **Parameters of the dataset.**

| Dataset | Fire | No fire | Total |
|---|---|---|---|
| Train | 9527 | 4157 | 13702 |
| Val | 2016 | 1007 | 3023 |
| Test | 2047 | 1126 | 3173 |
| Total | 13590 | 6038 | 19898 |

## 4.2 Evaluation metrics

In the experiment, the following metrics were used to evaluate the model: precision (P), recall (R), parameters, floating point operations (FLOPs), frames per second (FPS), mAP@0.5, and mAP@0.5:0.95. The calculation formulas are as follows:

$$Precision = \frac{TP}{TP + FP} \tag{17}$$

$$Recall = \frac{TP}{TP + FN} \tag{18}$$

$$FPS = \frac{Frames}{Time} \tag{19}$$

$$AP = \int_0^1 (Precision \times Recall)dx \tag{20}$$

$$mAP = \frac{1}{n} \sum_{i=1}^{n} AP_i \tag{21}$$

Specifically, TP (True Positive) represents the number of samples correctly predicted as positive, FP (False Positive) represents the number of samples predicted as positive but actually negative, and FN (False Negative) represents the number of samples predicted as negative but actually positive. The parameters metric reflects the total number of model parameters, indicating the model's size and complexity. FLOPs measure the number of operations required for one forward inference pass, offering an estimate of the computational efficiency and resource demand. FPS indicates the number of image frames the model can process in one second, providing insight into its real-time processing capability and overall responsiveness during deployment.

In Eq (21), n represents the number of classes. When n = 1, as is the case in this experiment focusing solely on the "fire" class, the mAP (mean Average Precision) value is equivalent to the AP (Average Precision) for this task. Additionally, mAP@0.5 refers to the mean average precision at an IoU (Intersection over Union) threshold of 0.5, which measures the detection accuracy at this specific overlap between predicted and ground-truth bounding boxes. Meanwhile, mAP@ 0.5:0.95 calculates the average precision across a range of IoU thresholds, from 0.5 to 0.95, with a step size of 0.05, providing a more comprehensive assessment of the model's performance by accounting for different degrees of overlap.

## 4.3 Experimental results and analysis on PC

The FCMI-YOLO algorithm was first trained and validated on a PC. Subsequently, the trained model was converted to the RKNN format using the RKNN-Toolkit2 and deployed onto edge devices for further validation and evaluation.

**4.3.1 Experimental environment.** The training environment is detailed in Table 3, and the main training parameters for the fire detection model are listed in Table 4. The experiment utilizes the pre-trained weights of YOLOv5s and employs the SGD optimizer for training the model.

**4.3.2 Performance comparison of different YOLOv5 model versions.** Fast and accurate fire detection is crucial for fire source control and early detection during the ignition stage. As shown in Table 5, there are significant performance differences between different versions of the YOLOv5 model in terms of detection accuracy and inference speed. YOL Ov5s strikes a good balance between accuracy (87.2%) and inference speed (98 FPS), while maintaining a

**Table 3. Model train environment.**

| Test Environment | Details |
|---|---|
| Operating system | Ubuntu22.04 |
| CPU | Xeon E5-2650 v4 |
| GPU | NVIDIA GeForce RTX 3060 12G |
| Programming language | Python3.8 |
| Deep learning framework | Pytroch2.0.0 |

**Table 4. Primary training parameters for the model.**

| Test Environment | Details |
|---|---|
| Epochs | 200 |
| Batch size | 16 |
| Image size | $640 \times 640$ |
| Optimization algorithm | SGD |
| Initial learning rate | 0.01 |
| Weight decay | 0.0005 |
| Momentum | 0.937 |

**Table 5. Performance of fire detector based on different model versions of YOLOv5.**

| Model | P(%) | FLOPs(G) | FPS |
|---|---|---|---|
| YOLOv5n | 83.2 | 4.6 | 105.6 |
| YOLOv5s | 87.2 | 15.9 | 98.8 |
| YOLOv5m | 85.6 | 49.2 | 80.1 |
| YOLOv5l | 89.0 | 109.6 | 56.3 |

low FLOPs count (15.9M), making it suitable for reliable detection performance with lower computational cost on resource-constrained edge devices. In contrast, although YOLOv5n achieves the fastest inference speed (105 FPS) and the lowest FLOPs (4.6M), its mAP is only 83.2%, which is insufficient for meeting the accuracy requirements in fire detection. On the other hand, YOLOv5l demonstrates the best detection accuracy (89.0% mAP), but its relatively slow inference speed (56.3 FPS) and large FLOPs (109.6M) limit its application on edge devices. Overall, YOLOv5s, with its excellent performance and efficient computational capability, emerges as the preferred model for edge-based fire detection tasks and has been selected as the base network for this study.

**4.3.3 FasterNext comparison experiment.** To evaluate the performance of the FasterNext module, the C3 modules in the backbone, neck, and head networks of YOLOv5s were replaced with FasterNext modules, and their performance was compared on the constructed fire dataset. As shown in Table 6, when only replacing the C3 module in the backbone network, the model's precision increased by 1.9% to 89.1%, recall improved by 4.7% to 83.4%, and mAP@0.5 increased by 1.3% to 87.8%. Furthermore, as illustrated in Fig 9, the model demonstrated greater robustness under various challenging conditions, such as daytime, nighttime, rain, and fog. These improvements primarily originate from two key innovations in FasterNext. First, using PConv convolution reduces computational costs to 1/16 of conventional convolutions and significantly enhances computational efficiency through optimized memory access. Second, adopting the SiLU activation function ensures that normalization and activation are sequentially applied after each PWConv, constructing a more refined non-linear feature transformation pathway while mitigating potential feature loss issues associated with the C3 module. Although the inference speed slightly decreased from 98.8 FPS to 93.3

**Table 6. Performance comparison of YOLOv5s with FasterNext module replacement.**

| Backbone | Neck | P(%) | R(%) | mAP@0.5 | mAP@0.5:0.95 | Parameters(M) | FLOPs(M) | FPS |
|---|---|---|---|---|---|---|---|---|
| C3 | C3 | 87.2 | 78.7 | 86.5 | 47.8 | 7.0 | 15946 | 98.8 |
| FasterNext | C3 | 89.1 | 83.4 | 87.8 | 48.4 | 6.3 | 13836 | 93.3 |
| C3 | FasterNext | 89.1 | 84.5 | 87.2 | 46.5 | 6.4 | 14740 | 98.2 |
| FasterNext | FasterNext | 88.5 | 83.7 | 86.9 | 46.4 | 5.7 | 12630 | 88.9 |

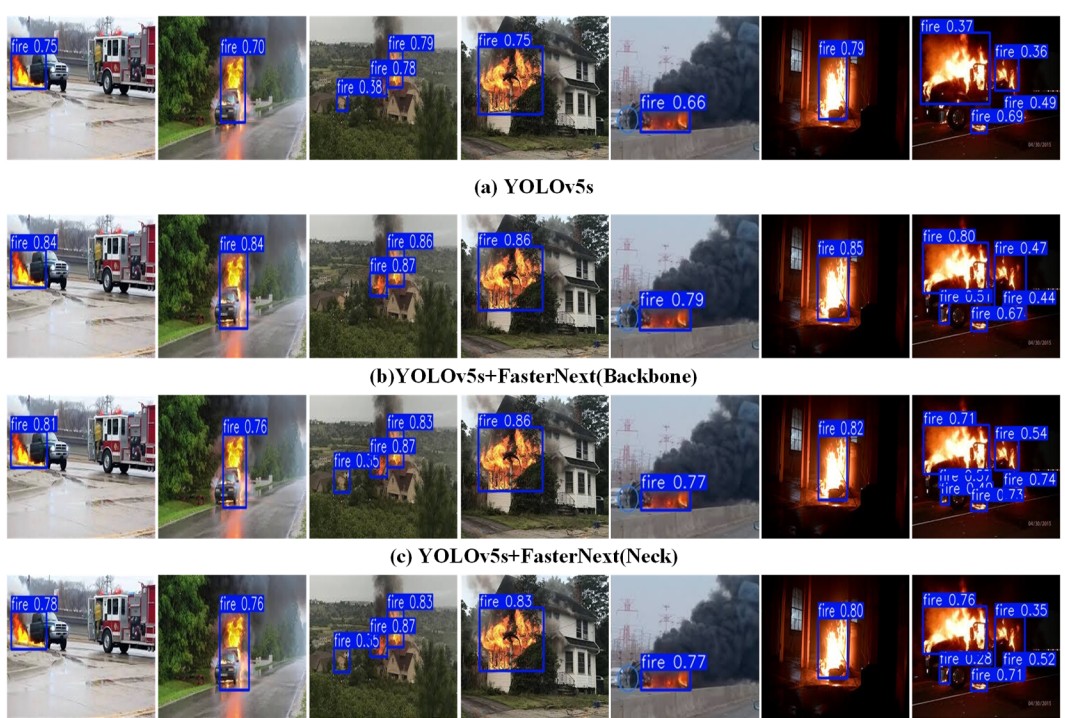

**Fig 9. Comparison of detection results of different methods in FasterNext.** (a) YOLOv5s. (b) YOLOv5s + FasterNext (Backbone). (c) YOLOv5s + FasterNext(Neck). (d) YOLOv5s + FasterNext(Backbone + Neck).

FPS, FasterNext achieves a better balance between accuracy improvement and computational efficiency, making it particularly suitable for complex object detection tasks in resource-constrained environments. Experimental results indicate that FasterNext not only overcomes the primary limitations of the C3 module but also achieves comprehensive performance enhancements through structural innovations.

**4.3.4 Loss function comparison experiment.** Based on the introduction of the FasterNext module, CCFM, and MLCA mechanism, this experiment adjusts the *ratio* of Inner-DIoU within the range of [0.6, 1.0]. According to Fig 10, the experimental results indicate that the model achieves optimal performance when the ratio is set to 0.8, with a mAP@0.5 reaching 88.0%.

To evaluate the detection performance of the proposed Inner-DIoU loss function, comprehensive comparative experiments were conducted on the constructed fire dataset. As shown in Table 7, a comparison of DIoU with benchmark methods such as CIoU, DIoU, EIoU, and GIoU along with their Inner and Focal variants [49], reveals that Inner-DIoU achieved the best performance in terms of mAP@0.5 and recall. Specifically, Inner-DIoU

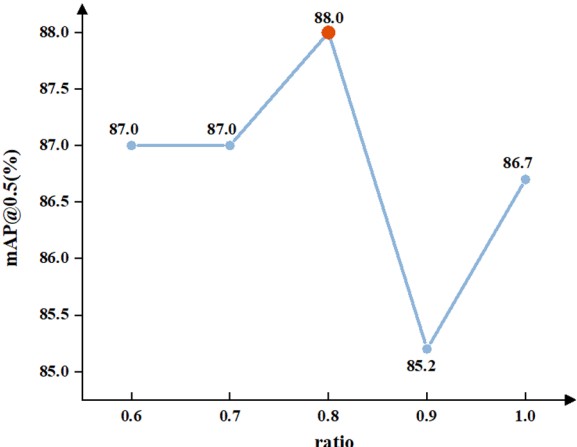

**Fig 10. Comparison of mAP@0.5 for different ratios.**

**Table 7. Performance comparison of different loss functions.**

| Loss Function | mAP@0.5 (%) | R (%) |
|---|---|---|
| CIoU | 87.3 | 82.4 |
| DIoU | 87.3 | 82.9 |
| EIoU | 86.8 | 81.4 |
| GIoU | 87.5 | 81.1 |
| Focal-CIoU | 86.3 | 85.0 |
| Focal-DIoU | 87.5 | 83.9 |
| Focal-EIoU | 87.0 | 84.0 |
| Focal-GIoU | 87.0 | 83.3 |
| Inner-CIoU | 86.6 | 82.2 |
| Inner-DIoU | 88.0 | 84.4 |
| Inner-EIoU | 85.2 | 78.8 |
| Inner-GIoU | 87.1 | 83.7 |

attained a mAP@0.5 of 88.0% and a recall of 84.4%, representing improvements of 0.7% and 2.0%, respectively, over the baseline CIoU. Experimental results demonstrate that Inner-DIoU dynamically adjusts the scaling ratio of the auxiliary bounding box based on the actual scale of fire targets, effectively enhancing the model's ability to perceive sudden variations in fire target sizes.

**4.3.5 Ablation experiments.** In this ablation experiment, each improvement stage of FCMI-YOLO was evaluated to assess its effectiveness in fire detection tasks. The "✓" symbol indicates the incorporation of a specific method.

As shown in Table 8, introducing each module individually results in a certain improvement in mAP@0.5, with the FasterNext module achieving the most significant increase of 1.3%. FasterNext, by incorporating a lightweight structure combining PConv and PWConv, reduces redundant computations while enhancing feature extraction capabilities, enabling the network to extract more discriminative fire features with a lower computational cost. However, since the PConv and PWConv combination primarily focuses on the central region of the input, it may have limitations in capturing edge details of fire targets, which could lead to a slight decrease in recall. Additionally, the module significantly reduces the number of model parameters and FLOPs, making it more suitable for edge computing environments.

**Table 8. Ablation experiments results of YOLOv5s.**

| FasterNext | CCFM | MLCA | Inner-DIoU | P | R | mAP@0.5 | Parameters | FLOPs (M) | FPS |
|---|---|---|---|---|---|---|---|---|---|
| | | | | 87.2 | 78.7 | 86.5 | 7.0 | 15946 | 98.8 |
| ✓ | | | | 89.1 | 83.4 | 87.8 | 6.3 | 13836 | 93.3 |
| | ✓ | | | 88.7 | 82.3 | 86.9 | 4.9 | 13390 | 96.1 |
| | | ✓ | | 86.7 | 83.4 | 87.1 | 7.0 | 15984 | 98.0 |
| | | | ✓ | 89.2 | 81.9 | 87.4 | 7.0 | 15946 | 99.2 |
| | ✓ | ✓ | | 87.2 | 83.4 | 86.6 | 4.9 | 13428 | 95.3 |
| ✓ | ✓ | | | 86.7 | 81.9 | 85.9 | 4.2 | 11280 | 96.0 |
| ✓ | ✓ | ✓ | | 88.2 | 82.4 | 87.8 | 4.2 | 11299 | 90.8 |
| ✓ | ✓ | ✓ | ✓ | 88.4 | 84.4 | 88.0 | 4.2 | 11299 | 91.2 |

Introducing the CCFM and MLCA mechanisms in the YOLOv5s neck network resulted in a 4.7% improvement in recall and a noticeable reduction in false positives, especially in scenarios with small fire targets or complex backgrounds, where the detection performance became more stable. This improvement is largely due to CCFM enhancing the interaction between high and low-level features through cascaded feature fusion, which improves the representation of small fire targets. Meanwhile, MLCA, by combining spatial and channel information, allows the network to focus more effectively on key fire features while suppressing background noise. Furthermore, this optimization reduced the model parameters by 30% and FLOPs by 15.9%, demonstrating higher computational efficiency that meets the stringent resource constraints of edge devices.

After introducing the Inner-DIoU loss function in YOLOv5s, the model's precision, recall, and mAP@0.5 were improved by 2%, 3.2%, and 0.9%, respectively. This improvement dynamically adjusts the scaling parameters of the auxiliary bounding box, effectively enhancing the network's adaptability to changes in the scale of fire targets, particularly in scenarios with significant variations in target size, where recall benefits were particularly evident. Inner-DIoU, by reconstructing the bounding box regression mechanism, strengthens the model's ability to capture multi-scale fire features, improving target differentiation while alleviating the bounding box localization bias caused by scale sensitivity in traditional methods, thus achieving a synergistic optimization of model robustness and detection accuracy.

When FasterNext, CCFM, MLCA, and Inner-DIoU are combined, the model achieves an optimal balance. The number of parameters is reduced by 40%, and the floating-point operations are reduced by 29%. While maintaining real-time performance, mAP@0.5 increases by 1.5% and recall improves by 5.7%. The experiments demonstrate that FCMI-YOLO achieves the best balance between accuracy and inference speed, meeting the stringent requirements of edge devices for high-precision, fast-response, and resource-efficient fire detection, providing essential support for efficient real-time fire monitoring on edge platforms.

**4.3.6 Visualization analysis.** To comprehensively verify the performance of the FCMI-YOLO model in fire detection tasks, visualization experiments were conducted under different exposure conditions, including normal exposure, underexposure, and overexposure, simulating complex scenarios that may arise in practical applications.

In the experiments, a detailed comparison was conducted between the FCMI-YOLO and the YOLOv5s. According to Fig 11, the FCMI-YOLO model not only accurately detects fire sources at close range (Fig 11 (**a**)), at long distances (Fig 11 (**b**)), and in nighttime environments (Fig 11 (**c**)), but also outperforms the YOLOv5s model in accuracy across all scenarios. Notably, under nighttime conditions, the YOLOv5s model tends to exhibit missed detections and decreased accuracy when detecting multiple or small fire targets as the exposure

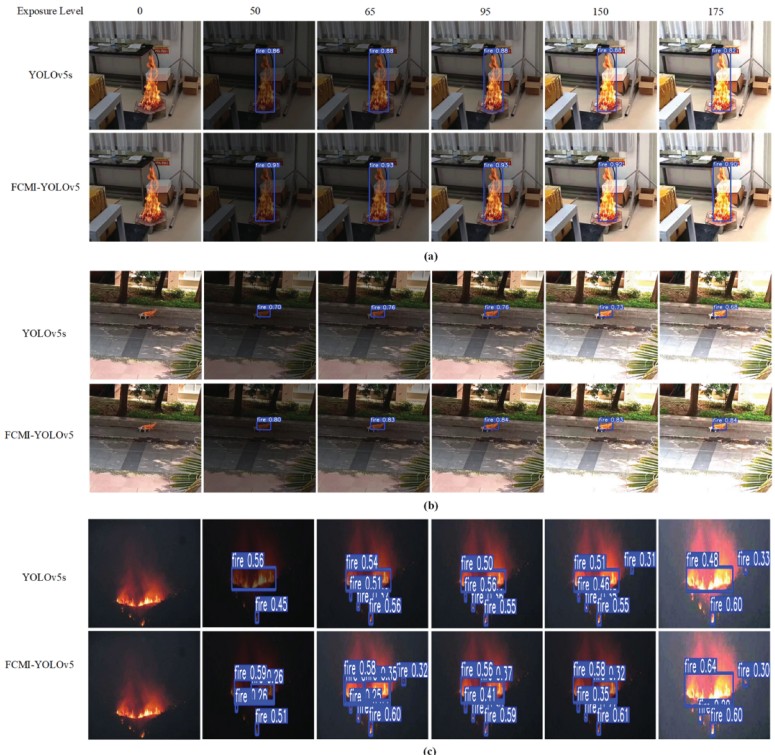

**Fig 11. Detection results of YOLOv5s and FCMI-YOLO under different exposure levels.** (a) Fire in the close interior. (b) Remote outdoor fire. (c) Fire in the Night.

levels fluctuate. In contrast, FCMI-YOLO effectively reduces missed detections, demonstrates higher accuracy in small-object detection, and improves overall detection precision, exhibiting strong anti-interference capability and robustness.

Overall, the FCMI-YOLO model demonstrates significant advantages in managing complex scenarios in fire detection tasks. Its sensitivity and accuracy under various exposure conditions provide more reliable technical support for practical applications in fire warning and monitoring.

**4.3.7 Comparison of mainstream algorithms.** To further evaluate the proposed algorithm, a comparative analysis with mainstream methods was conducted using the constructed fire dataset. As shown in Fig 12, FCMI-YOLO achieves the highest recall rate (84.4%) among all models, demonstrating strong adaptability in complex environments. This high recall is particularly important for fire safety applications, where missed detections may delay early warning and compromise response in industrial or long-range monitoring scenarios.

As shown in Table 9, compared to the proposed FCMI-YOLO, the parameters and flops of Faster R-CNN, SSD, PP-YOLOEs, YOLOv3, YOLOv4, YOLOv5s, YOLOv6s, YOLOv7, YOLOv7-Tiny, YOLOv8s, YOLOv8s-World, YOLOv8s-FasterNet, YOLOv9s, YOLOv10s, YOLOv11s, YOLOv11s-MobileNetv4, and YOLOv11s-EMO are 32.55, 5.62, 1.88, 14.86, 15.33, 1.67, 4.12, 8.86, 1.43, 2.64, 3.19, 2.05, 2.36, 1.93, 2.24, 1.24, and 2.02 times larger, respectively. The flops of these models are 32.72, 15.47, 1.54, 5.83, 5.36, 1.41, 3.90, 9.30, 1.15, 2.53, 2.88, 1.93, 3.59, 2.19, 1.90, 0.93, and 1.02 higher than FCMI-YOLO, respectively. Moreover, FCMI-YOLO achieves a balanced performance with 88.0% mAP@0.5 and 91.2 FPS, surpassing most mainstream algorithms in inference speed and accuracy. While its precision

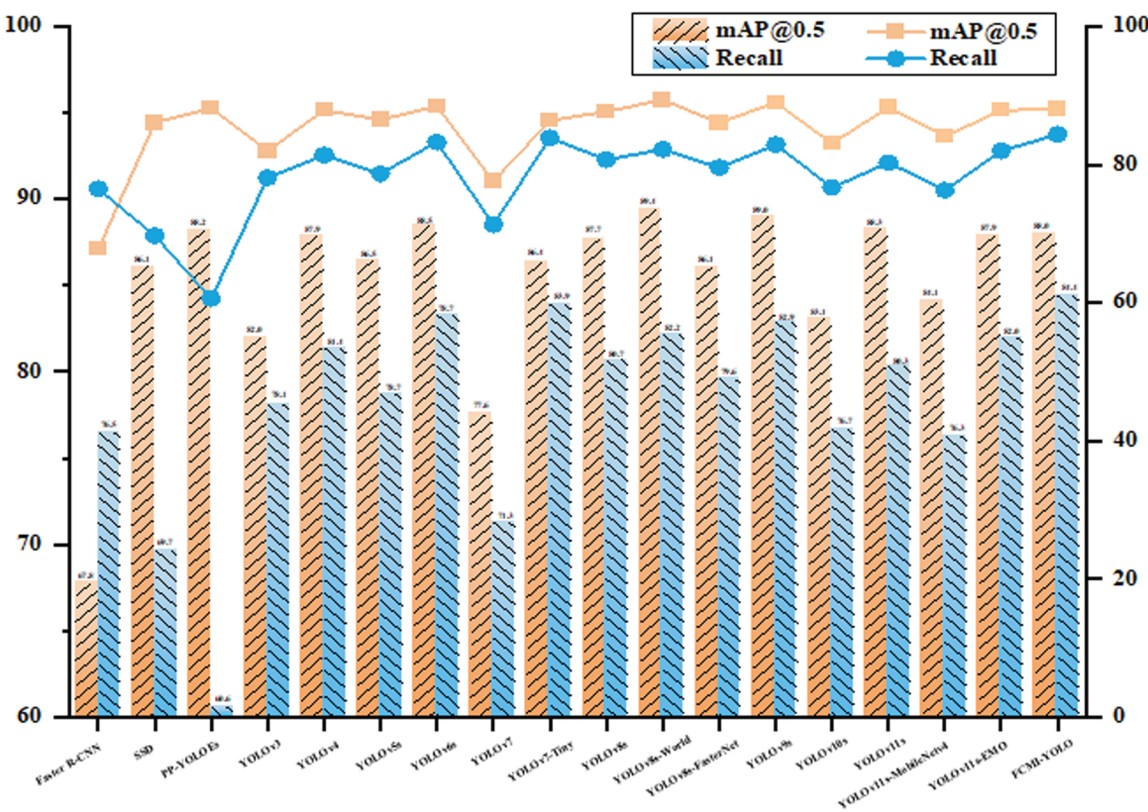

**Fig 12. Comparison of mAP@0.5 and Recall for mainstream algorithms.**

slightly trails behind PP-YOLOEs (88.2%), YOLOv6s (88.5%), YOLOv8s-World (89.4%), YOLOv9s (89.0%), and YOLOv11s (88.3%), these models exhibit significantly higher computational costs that hinder embedded deployment. Specifically, YOLOv8s-World requires 13.4M parameters, more than three times the 4.2M parameters of FCMI-YOLO, and 32.6G flops, nearly three times higher than FCMI-YOLO's 11.3G flops. However, it delivers only 36.3 FPS, 60.1% slower than our model. Similarly, YOLOv6s achieves 77.8 FPS with 17.3M parameters, over four times that of FCMI-YOLO, and 44.1G flops, almost four times higher, YOLOv9s attains 44.2 FPS, 51.5% slower despite having 9.9M parameters, about two and a half times that of FCMI-YOLO, and 40.6G flops, nearly four times greater.

To intuitively verify the effectiveness of FCMI-YOLO, a visual comparison analysis was conducted using the four optimal models from Table 9. As shown in Fig 13, under challenging conditions such as strong light, occlusion and long-distance scenarios, YOLOv6, YOLOv9, and YOLOv11 exhibit varying degrees of missed detections. In contrast, FCMI-YOLO consistently achieves the highest detection accuracy across different test scenarios.

In summary, FCMI-YOLO demonstrates a distinct advantage over mainstream detection algorithms by achieving an optimal balance between precision, inference speed, and model lightweight. With only 4.2M parameters and 11.3 GFLOPs, it significantly reduces computational overhead while delivering competitive accuracy and recall rates. This enables deployment on edge devices without sacrificing detection quality. Unlike larger models such as YOLOv8s-World and YOLOv9s, which require over twice the parameters and FLOPs while operating at much lower FPS, FCMI-YOLO offers both real-time responsiveness and robust

**Table 9. Performance comparison of mainstream algorithms.**

| Model | Parameters(M) | FLOPs(G) | Recall(%) | mAP@0.5(%) | FPS |
|---|---|---|---|---|---|
| Faster R-CNN | 136.7 | 369.7 | 76.5 | 67.8 | 13.5 |
| SSD | 23.6 | 174.8 | 69.7 | 86.1 | 42.3 |
| PP-YOLOEs [50] | 7.9 | 17.36 | 60.6 | 88.2 | 86.4 |
| YOLOv3 [51] | 62.4 | 65.9 | 78.1 | 82.0 | 28.9 |
| YOLOv4 [52] | 64.4 | 60.6 | 81.4 | 87.9 | 24.3 |
| YOLOv5s | 7.0 | 15.9 | 78.7 | 86.5 | 98.8 |
| YOLOv6s [53] | 17.3 | 44.1 | 83.3 | 88.5 | 77.8 |
| YOLOv7 [54] | 37.2 | 105.1 | 71.3 | 77.6 | 78.3 |
| YOLOv7-Tiny [55] | 6.0 | 13.0 | 83.9 | 86.4 | 66.1 |
| YOLOv8s | 11.1 | 28.6 | 80.7 | 87.7 | 84.0 |
| YOLOv8s-World [56] | 13.4 | 32.6 | 82.2 | 89.4 | 36.3 |
| YOLOv8s-FasterNet [42] | 8.6 | 21.8 | 79.6 | 86.1 | 78.4 |
| YOLOv9s [57] | 9.9 | 40.6 | 82.9 | 89.0 | 44.2 |
| YOLOv10s [58] | 8.1 | 24.8 | 76.7 | 83.1 | 48.1 |
| YOLOv11s [59] | 9.4 | 21.5 | 80.3 | 88.3 | 42.5 |
| YOLOv11s-MobileNetv4 [60] | 5.2 | 10.5 | 76.3 | 84.1 | 76.0 |
| YOLOv11s-EMO [61] | 8.5 | 11.5 | 82.0 | 87.9 | 31.5 |
| FCMI-YOLO | 4.2 | 11.3 | 84.4 | 88.0 | 91.2 |

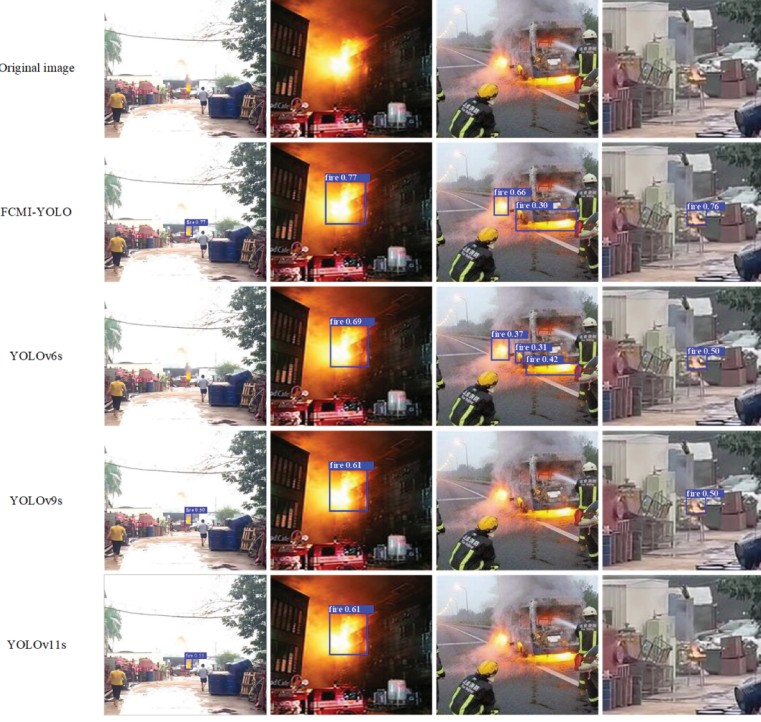

**Fig 13. Detection results of FCMI-YOLO, YOLOv6s, YOLOv9s, and YOLOv11s.**

detection performance. Furthermore, its superior recall capability is particularly valuable for safety-critical applications, where failure to detect early-stage fires could lead to uncontrollable spread and escalated hazards. These results affirm the practical applicability and technical strength of the proposed method in resource-constrained fire detection scenarios.

## 4.4 Model deployment

**4.4.1 Hardware specifications and environment configuration.** The experiment utilizes the Orange Pi 5 Plus as the edge device testing platform. According to Fig 14, the system is primarily composed of five components: the image acquisition module, image processing module, display module, network communication module, and storage module. The platform is powered by the Rockchip RK3588 processor, which integrates a Neural Network Processing Unit (NPU) with a computational power of 6 TOPS. The system uses an IMX577 camera to capture images.

First, the image acquisition module receives video stream data from an external source through the IMX577 camera and transmits it to the image processing module, where it is temporarily stored in LPDDR4 memory. Next, the RK3588 processor utilizes the CPU and GPU in the multimedia processing unit to perform image enhancement on the video stream, runs the object detection algorithm, and accelerates inference speed using the NPU. Finally, the detection results are optimized through video stream compression and encoding, then transmitted in real-time to the display module via the HDMI interface for displaying the detection results.

**4.4.2 Asynchronous multi-threaded processing.** In detection tasks, the Orange Pi 5 Plus leverages NPU acceleration, with its utilization directly affecting inference speed. To maximize NPU performance, this study employs an asynchronous multithreading method, which allows multiple tasks to be executed in parallel, thereby reducing idle time and improving resource utilization efficiency. In this method, asynchronous operations, such as submitting inference requests to the NPU, are initiated by the main thread without waiting for their completion. This non-blocking mechanism ensures that the main thread remains responsive and can continue handling other tasks, such as preprocessing input frames or managing communication with peripheral devices. Meanwhile, worker threads (or background threads) are responsible for executing time-consuming inference operations. Once the NPU completes an inference task, the results are passed back to the main thread, enabling prompt postprocessing and visualization. As shown in Table 10, the asynchronous multithreading method significantly improves overall NPU utilization by enabling concurrent task execution and

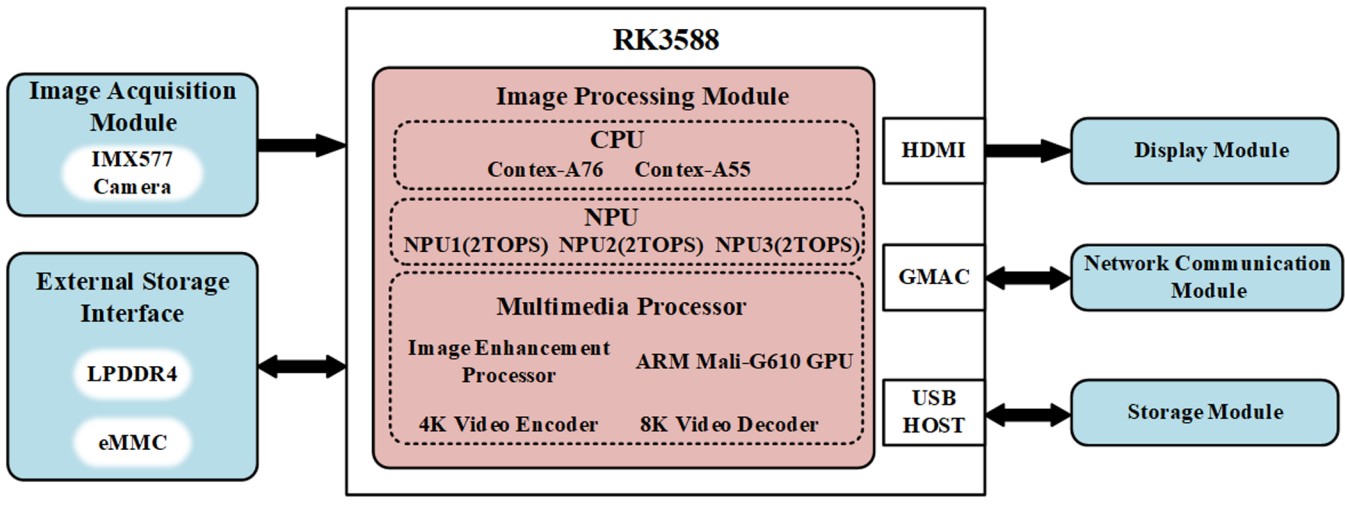

**Fig 14. System diagram.**

**Table 10. NPU utilization and FPS Under different processing methods.**

| Embedded Device | Method | NPU1 (%) | NPU2 (%) | NPU3 (%) | FPS |
|---|---|---|---|---|---|
| OrangePi 5 Plus | Bare Metal | 45 | 0 | 0 | 5.5 |
| OrangePi 5 Plus | Asynchronous Multithreading | 85 | 83 | 84 | 23.4 |

more balanced resource scheduling. Consequently, the system's frame rate (FPS) increased from 5.5 to 23.4, demonstrating substantial improvements in responsiveness and real-time performance.

**4.4.3 Deployment results.** To evaluate the detection performance of the FCMI-YOLO algorithm on edge devices, experiments were conducted on the Orange Pi 5 Plus using the constructed fire dataset. As shown in Fig 15, the algorithm achieved a mAP of 81.45%, demonstrating excellent detection accuracy.

Additionally, outdoor fire detection experiments were conducted to further assess the real-world performance of the proposed algorithm on resource-constrained edge devices. As shown in Fig 16, experiments were performed at distances of 30 m and 75 m from the fire source, with FCMI-YOLO and YOLOv5s deployed on the OrangePi 5 Plus. At 30 m, FCMI-YOLO achieved a precision of 88% and an inference speed of 23.4 FPS, outperforming YOLOv5s, which recorded 72% precision at 25.8 FPS. At 75 m, FCMI-YOLO maintained a precision of 70% with the same FPS, while YOLOv5s dropped to 59% precision. Although FCMI-YOLO operates at a slightly lower FPS, it consistently delivers higher detection accuracy, especially under long-range and low-resolution fire conditions. These results indicate that FCMI-YOLO demonstrates superior robustness and detection stability compared to YOLOv5s in outdoor environments, validating its suitability for real-time fire detection on edge devices.

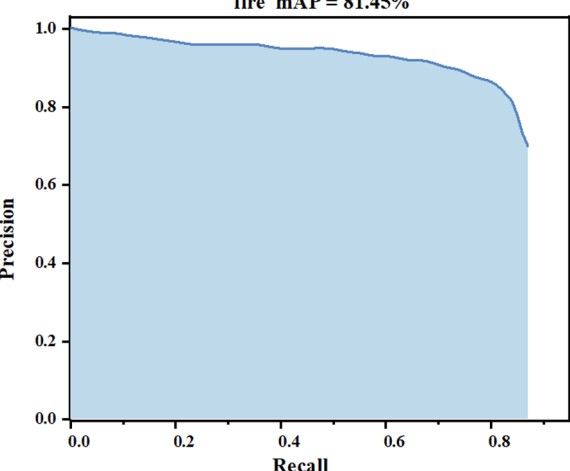

**Fig 15. mAP performance of FCMI-YOLO on the Orange Pi 5 Plus.**

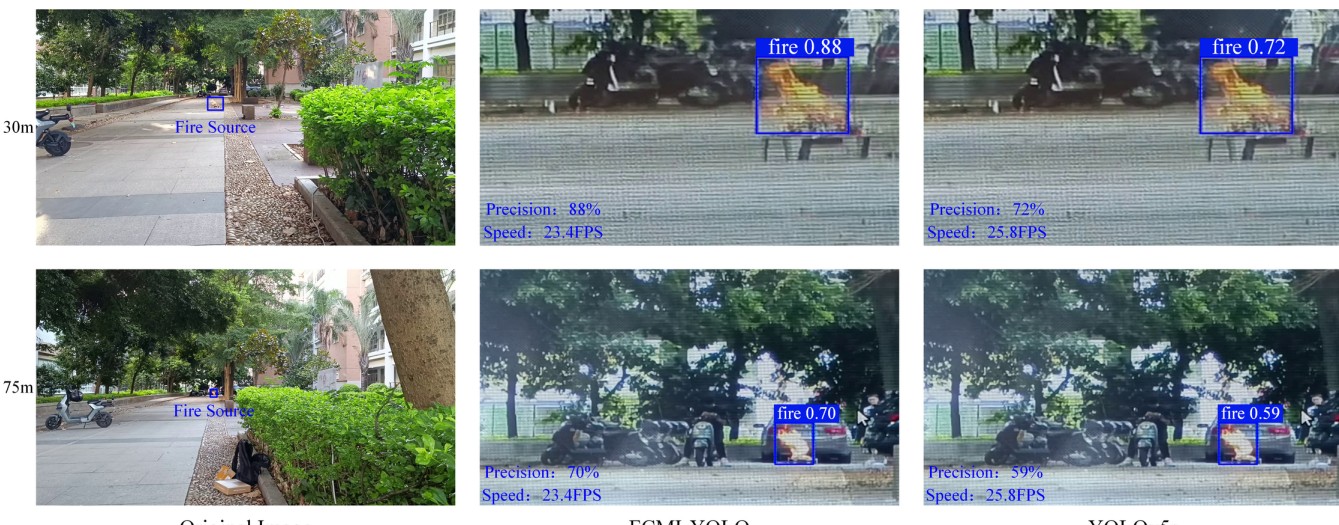

**Fig 16. Detection results of FCMI-YOLO and YOLOv5s at 30m and 75m on the OrangePi 5 Plus.**

## 5 Conclusion

To achieve the optimal trade-off between accuracy and inference speed for fire detection, a realtime fire detection algorithm, FCMI-YOLO, is proposed and successfully deployed on edge devices for realtime detection tasks involving fire images, videos, and camera streams. First, a light-weight feature extraction module, FasterNext is introduced, which combines PConv and PWC-onv to reduce model complexity while incorporating the nonlinear activation function SiLU to enhance fire feature representation. Second, the CCFM and MLCA mechanisms are integrated to improve recall through hierarchical feature fusion and the utilization of spatial and channel information. Finally, the Inner-DIoU loss function is proposed, introducing an auxiliary bounding box constraint mechanism to optimize multi-scale object localization and enhance scale variation awareness. Experimental results demonstrate that FCMI-YOLO achieves 88.0% mAP@0.5 and 91.2 FPS on a PC, exhibiting significant advantages over other YOLO variants regarding model parameters, detection accuracy, and inference speed. When deployed on an edge device, it maintains an mAP of 81.45% and a realtime performance of 23.4 FPS, providing an efficient solution for real-time fire monitoring.

However, the proposed algorithm still presents several limitations. First, in environments with strong wind or reflective surfaces, fire deformation, and mirror reflection of fire may compromise feature extraction and increase false positives, reducing detection accuracy. Second, although the model has been lightweighted to achieve real-time detection on edge devices, there is still room for further improvement in inference speed for practical applications. Lastly, the performance on edge devices varies depending on the hardware, making model optimization for different platforms a challenge for future work. Future research will focus on enhancing the generalization ability of FCMI-YOLO to adapt to a wider range of detection scenarios. Additionally, further lightweight strategies will be explored to improve detection speed regarding FPS.

## Author contributions

**Conceptualization:** Junjie Lu, Yuchen Zheng.

**Data curation:** Junyan Zhang.

**Formal analysis:** Wenzao Shi.

**Funding acquisition:** Wenzao Shi, Yunping Wu.

**Investigation:** Yunping Wu.

**Methodology:** Liwei Guan.

**Project administration:** Liwei Guan, Wenzao Shi, Yunping Wu.

**Resources:** Yuchen Zheng.

**Supervision:** Liwei Guan.

**Validation:** Yuchen Zheng.

**Writing – original draft:** Junjie Lu, Yuchen Zheng.

**Writing – review & editing:** Liwei Guan, Bing Lin, Yunping Wu.

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
