## [Decision Letter · Decision Letter 0]

26 Feb 2025

PONE-D-25-01361FCMI-YOLO: A Deep Learning-Based Algorithm for Real-Time Fire Detection on Edge DevicesPLOS ONE

Dear Dr. Guan,

Thank you for submitting your manuscript to PLOS ONE. After careful consideration, we feel that it has merit but does not fully meet PLOS ONE’s publication criteria as it currently stands. Therefore, we invite you to submit a revised version of the manuscript that addresses the points raised during the review process. Based on the reviewers' comments, a resubmission with major revisions is proposed. Please provide a detailed response to each of these comments.

We look forward to receiving your revised manuscript.

Kind regards,

Irving A. Cruz-Albarran

Academic Editor

PLOS ONE

Journal Requirements:

3. In the online submission form, you indicated that the data cannot be shared publicly because they involve subsequent applications for patents, software copyright, and the publication of project deliverables. The data underlying the results presented in this study are available upon request from the corresponding author.

Additional Editor Comments:

Please provide a clear and detailed response to each of the reviewer's comments. 

Reviewers' comments:

Reviewer's Responses to Questions

**Comments to the Author**

1. Is the manuscript technically sound, and do the data support the conclusions?

Reviewer #1: Partly

Reviewer #2: No

2. Has the statistical analysis been performed appropriately and rigorously? 

Reviewer #1: Yes

Reviewer #2: No

3. Have the authors made all data underlying the findings in their manuscript fully available?

Reviewer #1: Yes

Reviewer #2: No

4. Is the manuscript presented in an intelligible fashion and written in standard English?

Reviewer #1: No

Reviewer #2: No

5. Review Comments to the Author

Reviewer #1: 1- The authors should emphasize the merits of the developed method.

2-in the section simulation results, the proposed approach should compare with other relevant algorithms, the authors are asked to further provide comparisons to better show the advantages of their approach.

3. the literature review is not good enough.

4.some minor grammatical errors should be eliminated .the authors are kindly asked to double check the manuscript.

5.The abstract could be more concise and the point.

6.I suggest the authors consider exposes the equation more clearly so to achieve a broader audience.

7. pleas add more sentences to describe the limitations of this work. The advantages and disadvantages should be discussed.

Reviewer #2: 1- The title should be improved.

2- The objectives and the rationale of the study are recommended to be clearly stated.

3- The concluding remarks of the abstract are not well-written. It's merely the repetition of the objectives and title of the manuscript. Please add method limitations and justification to the abstract.

4- The innovation of using this study is not very clear. I do not see a clear reason that this study can perform better than others. Why did the authors choose the method for this study?

5- The necessity & novelty of the manuscript should be presented and stressed in the "Introduction" section.

6- The application/theory/method/study reported is not in sufficient detail to allow for its replicability and/or reproducibility. Therefore, it is suggested to make it clear to show all steps to build the model.

7- The problem statement and gap study are not clear.

8- The method is not clear. Therefore, it must be shown and clarified to show all steps.

9- The interpretation of results and study conclusions are not supported by providing the reasons behind why they show that. Therefore, it is recommended to deepen the discussion.

10- It is recommended to emphasize the strengths of the study clearly.

11- The limitations of the study should be stated.

12- The manuscript structure, flow, or writing needs some improvements.

13- The manuscript is benefit from language editing. The English of the paper is readable; however, I would suggest the authors to have it checked preferably by a native English-speaking person to avoid any mistakes.

14- I noticed that the conclusion section tends to repeat the abstract and results. The conclusion paragraph should be short, impactful, and direct the reader to this research's next steps and opportunities.

15- It will be nice to add some new references to show that your study is updated, such as: Zhou, Zhanxin, and Ruibo Wu. "Stock Price Prediction Model Based on Convolutional Neural Networks." Journal of Industrial Engineering and Applied Science 2.4 (2024): 1-7; Alakbari, Fahd Saeed, et al. "Prediction of critical total drawdown in sand production from gas wells: Machine learning approach." The Canadian Journal of Chemical Engineering 101.5 (2023): 2493-2509.; Alakbari, Fahd Saeed, et al. "Deep learning approach for robust prediction of reservoir bubble point pressure." ACS omega 6.33 (2021): 21499-21513.; Alakbari, Fahd Saeed, et al. "A gated recurrent unit model to predict Poisson's ratio using deep learning." Journal of Rock Mechanics and Geotechnical Engineering 16.1 (2024): 123-135; Zhou, Zhanxin, and Ruibo Wu. "Stock Price Prediction Model Based on Convolutional Neural Networks." Journal of Industrial Engineering and Applied Science 2.4 (2024): 1-7; Wu, Ruibo, Tao Zhang, and Feng Xu. "Cross-Market Arbitrage Strategies Based on Deep Learning." Academic Journal of Sociology and Management 2.4 (2024): 20-26.

6. PLOS authors have the option to publish the peer review history of their article (what does this mean?). If published, this will include your full peer review and any attached files.

Reviewer #1: No

Reviewer #2: No

---

## [Author Response · Author response to Decision Letter 1]

11 Apr 2025

Please refer to the Response to Reviewers document for detailed information.

Response to Reviewer 1’s Comments

Comment 1: The authors should emphasize the merits of the developed method.

Response: We appreciate the reviewer’s suggestion. To emphasize the advantages of the proposed method in this paper, we have taken the following measures:

First, we have emphasized the merits of the proposed FCMI-YOLO algorithm more clearly and thoroughly. Specifically, the following enhancements and advantages have been highlighted:

Algorithmic Improvements:

We designed the FasterNext backbone module, which replaces the original C3 module. This lightweight structure, composed of PConv and PWConv, significantly reduces parameters and computational cost while enhancing feature extraction capabilities in complex environments.

We integrated the Cross-Scale Feature Fusion Module (CCFM) and the Mixed Local Channel Attention (MLCA) mechanism into the neck network. These modules enhance the detection of small fire targets by improving the fusion of hierarchical features and enabling the model to focus on spatially and channel-wise important information.

We introduced the Inner-DIoU loss function, which enhances bounding box regression by adaptively adjusting auxiliary box scales, improving the model’s sensitivity to fire target size variations and irregular shapes.

Effectiveness of the Improvements:

As shown in the Ablation Experiments, our improvements led to a 1.5% increase in mAP@0.5, a 5.7% improvement in recall, and a 40% reduction in parameters compared to the YOLOv5s baseline. Furthermore, FLOPs were reduced by 29%, and the algorithm maintained high-speed inference at 91.2 FPS on PC and 23.4 FPS on the Orange Pi 5 Plus edge device.

These results demonstrate that FCMI-YOLO effectively balances accuracy and computational efficiency, making it highly suitable for real-time fire detection in resource-constrained edge environments.

Furthermore, to comprehensively validate the overall performance of the proposed algorithm, we have systematically expanded Section 4.3.7, Comparison Experiments with Mainstream Algorithms. As shown in Table 1, six additional comparative experiments have been introduced, covering different application scenarios and evaluation dimensions. A detailed comparative analysis has been conducted based on multiple technical metrics, including Precision, FLOPs, and Recall. The experimental results demonstrate that, compared to existing mainstream algorithms, the proposed algorithm exhibits significant advantages in key performance indicators. These quantitative comparisons fully validate the breakthrough progress of our algorithm in achieving a balance between performance and efficiency, providing strong evidence of its superiority. To intuitively verify the effectiveness of FCMI-YOLO, a visual comparison analysis was conducted using the four optimal models from Table 1. As shown in Fig 1, under challenging conditions such as strong light, occlusion, and long-distance scenarios, YOLOv6, YOLOv9, and YOLOv11 exhibit varying degrees of missed detections. In contrast, FCMI-YOLO consistently achieves the highest detection accuracy across different test scenarios.

[1] Cao X, Su Y, Geng X, Wang Y. YOLO-SF: YOLO for Fire Segmentation Detection. Ieee Access. 2023;11:111079-111092.

[2] Wang S, Wu M, Wei X, Song X, Wang Q, Jiang Y, et al. An advanced multi-source data fusion method utilizing deep learning techniques for fire detection. Engineering Applications of Artificial Intelligence. 2025;142.

[3] He H, Zhang Z, Jia Q, Huang L, Cheng Y, Chen B. Wildfire detection for transmission line based on improved lightweight YOLO. Energy Reports. 2023;9:512-20.

Comment 2: In the section simulation results, the proposed approach should compare with other relevant algorithms, the authors are asked to further provide comparisons to better show the advantages of their approach.

Response: We appreciate the reviewer’s suggestion. To systematically evaluate the overall performance of the proposed algorithm, we have significantly expanded Section 4.3.7, Comparison Experiments with Mainstream Algorithms. This expansion incorporates six representative advanced object detection algorithms as benchmark references, including PP-YOLOEs [1], YOLOv7-Tiny [2], YOLOv8s-World [3], YOLOv8s-FasterNet [4], YOLOv11s-MobileNetv4 [5], and YOLOv11s-EMO [6], as shown in Table 1.

To further highlight the advantages of the proposed method, we have added visualization experiments comparing four of the most competitive advanced object detection algorithms (YOLOv6, YOLOv9, YOLOv11, and FCMI-YOLO), as shown in Figure 2. These visual comparison results provide a more comprehensive demonstration of the innovative contributions and technical breakthroughs of this study.

[1] Xu S, Wang X, Lv W, Chang Q, Cui C, Deng K, et al. PP-YOLOE: An evolved version of YOLO. arXiv preprint arXiv:2203.16250. 2022.

[2] Zhu J, Zhang J, Wang Y, Ge Y, Zhang Z, Zhang S. Fire Detection in Ship Engine Rooms Based on Deep Learning. Sensors. 2023;23(14).

[3] Cheng T, Sone L, Ge Y, Liu W, Wang X, Shan Y, et al. YOLO-World: Real-Time OpenVocabulary Object Detection. IEEE/CVF Conference on Computer Vision and Pattern Recognition (CVPR); 2024.

[4] Chen J, Kao Sh, He H, Zhuo W, Wen S, Lee CH, et al. Run, don’t walk: chasing higher FLOPS for faster neural networks. Proceedings of the IEEE/CVF confer- ence on computer vision and pattern recognition; 2023.

[5] Qin D, Leichner C, Delakis M, Fornoni M, Luo S, Yang F, et al. MobileNetV4: Universal Models for the Mobile Ecosystem. arXiv preprint arXiv:2404.10518. 2024.

[6] Zhang J, Li X, Li J, Liu L, Xue Z, Zhang B, et al. Rethinking Mobile Block for Efficient Attention-based Models. arXiv preprint arXiv:2301.01146. 2023.

Comment 3: The literature review is not good enough.

Response: Thank you for your valuable feedback. We have carefully reviewed and revised the Related Work section to ensure that it provides a more comprehensive and detailed literature review. The citation distribution of the related works before and after the revision is illustrated in Figure 1, where the references highlighted in red indicate the newly added or revised citations, while those marked in blue represent the original references that have been retained. Specifically, we have expanded the discussion on existing fire detection algorithms and highlighted recent advancements in YOLO-based methods, as well as other relevant approaches. We have also emphasized the strengths and weaknesses of the methods to clearly establish the research gap that FCMI-YOLO aims to address.

In addition to the existing references, we have included studies that explore the optimization of YOLO models for fire detection in various environments, such as small fire targets, complex scenarios, and real-time deployment on edge devices. These references help contextualize the contributions of our work and demonstrate the advancements made in the field. Moreover, we have discussed the trade-off between accuracy and inference speed, as well as the challenges posed by computational complexity in real-time applications on resource-constrained devices.

To further strengthen the literature review, we have elaborated on the limitations of existing methods and how FCMI-YOLO improves upon them. Specifically, we have discussed how our method effectively balances accuracy and speed while minimizing the computational burden, making it particularly suitable for deployment on edge devices.

We believe these revisions provide a more complete and critical review of the relevant literature and better highlight the necessity and novelty of our proposed approach. Please refer to the updated Related Work section for further details.

Comment 4: Some minor grammatical errors should be eliminated .the authors are kindly asked to double check the manuscript.

Response: We sincerely appreciate your meticulous review of the manuscript's language quality. Our team members have conducted multiple rounds of cross-checking, focusing on correcting grammar, spelling, and sentence structure issues while ensuring consistency in the use of academic terminology. Throughout the revision process, we paid special attention to the accuracy of academic English expressions and the fluency of the writing, striving to meet the professional standards required by the journal. In the revised version, all language-related modifications have been clearly highlighted to facilitate review. Your valuable feedback has significantly contributed to improving the quality of our paper, for which we are deeply grateful. If you identify any further areas for improvement, we will fully cooperate in making the necessary revisions.

Comment 5: The abstract could be more concise and the point.

Response: Thank you for your valuable suggestion. In response, we have revised the Abstract to be more concise and focused on the key contributions of our work. The updated version highlights the core innovations of FCMI-YOLO and its performance improvements in a streamlined manner. Below is the revised version:

The rapid development of Internet of Things (IoT) technology and deep learning has propelled the deployment of vision-based fire detection algorithms on edge devices, significantly exacerbating the trade-off between accuracy and inference speed under hardware resource constraints. To address this issue, this paper proposes a real-time fire detection algorithm named FCMI-YOLO. Firstly, the FasterNext model, which is based on FasterNet, is introduced to reduce the computational of model parameters and enhance detection precision through lightweight design. Secondly, the Cross-Scale Feature Fusion Module (CCFM) and the Mixed Local Channel Attention (MLCA) mechanism are incorporated into the neck network, significantly improving detection performance for small fire targets and minimizing resource consumption. Finally, the Inner-DIoU loss function is proposed to optimize bounding box regression by incorporating auxiliary boxes. Experimental results on the fire dataset we constructed demonstrate that FCMI-YOLO increases mAP@50 by 1.5%, decreases the number of parameters by 40%, and lowers GFLOPs to 28.9% of those in YOLOv5s. These results demonstrate that FCMI-YOLO enhances fire detection accuracy while significantly reducing computational demands, making it highly valuable for widespread practical applications, particularly on resource-constrained edge devices.

If you have any further suggestions, we will continue to refine it.

Comment 6: I suggest the authors consider exposes the equation more clearly so to achieve a broader audience.

Response: We sincerely appreciate your valuable suggestions. Based on your feedback, we have comprehensively improved the presentation of formulas in our paper. In Section 3.2, FasterNext Network, to more intuitively demonstrate the advantages of the SiLU activation function, we have added a comparative formulation between ReLU and SiLU (as shown in Equations (1) and (2)). Through a formal mathematical expression, we illustrate their differences in key properties such as gradient continuity and output smoothness. Additionally, we incorporate experimental data to quantitatively analyze the improvements SiLU brings to model convergence speed and detection accuracy.

Additionally, in Section 3.4, Improved Loss Function, we have systematically derived the complete mathematical expression of the CIoU loss function (as shown in Equations (3)–(5)). Furthermore, we have conducted an in-depth analysis of key components such as the aspect ratio term, explaining their physical significance and their impact on the bounding box regression mechanism.

By presenting clearer mathematical expressions, we have enhanced the readability of the content, allowing readers from diverse academic backgrounds to better grasp our methodological innovations. Once again, we sincerely appreciate your constructive feedback, which is crucial to improving the quality of our paper. If further adjustments are needed, we will be happy to make the necessary refinements.

Comment 7: Please add more sentences to describe the limitations of this work. The advantages and disadvantages should be discussed.

Response: Thank you for your valuable feedback. We have expanded the discussion on the limitations of our research in the Conclusion section of the paper.

Currently, the algorithm’s detection accuracy needs improvement when dealing with complex environments such as distant fire incidents, strong winds, intense light, and reflective surfaces. Although the model has been optimized in terms of parameter size and computational cost, its inference speed still requires further enhancement to meet more stringent real-time application requirements. Additionally, due to hardware performance differences across edge devices, the algorithm's performance varies on different platforms, presenting new challenges for cross-platform optimization. To address these limitations, future research will focus on improving the generalization ability of FCMI-YOLO in diverse detection scenarios, exploring more efficient lightweight strategies to optimize detection speed, and developing deployment solutions that can adapt to different hardware platforms. These additional discussions will help readers more objectively assess the applicability and development potential of this research.

The updated content is as follows:

"First, its detection accuracy may decrease when fire targets are at long distances or in adverse environments, such as strong wind, intense light, or reflective surfaces. Second, although the model has been optimized for parameters and computational cost, there is still room for further improvement in inference speed for real-time applications. Lastly, the performance on edge devices varies depending on the hardware, making model optimization for different platforms a challenge for future work. Future research will focus on enhancing the generalization ability of FCMI-YOLO to adapt to a wider range of detection scenarios. Additionally, further lightweight strategies will be explored to improve detection speed regarding FPS. "

Response to Reviewer 2’s Comments

Comment 1: The title should be improved.

Response: Thank you for your suggestion. In response, we have revised the title to better reflect the efficiency aspect of our proposed method. The new title is: "FCMI-YOLO: An Efficient Deep Learning-Based Algorithm for Real-Time Fire Detection on Edge Devices."

This revision emphasizes the efficiency of the FCMI-YOLO algorithm, aligning with the core contributions of the paper.

Comment 2: The objectives and the rationale of the study are recommended to be clearly stated.

Response: Thank you for your valuable suggestion. In response, we have revised the manuscript to more clearly present the research objectives and the rationale behind our study, with a particular emphasis on the core challenges currently facing the field.

In the Abstract, we refined the background description to highlight the primary challenge this study addresses, the trade-off between accuracy and inference speed under limited computational resources. Specifically, we state: "The rapid development of Internet of Things (IoT) technology and deep learning has propelled the deployment of vision-based fire detection algorithms on edge devices, significantly exacerbating the trade-off between accuracy and inference speed under hardware resource constraints." This clearly defines the technical bottleneck that motivates our work.

In the Introduction, we strengthened the rationale by analyzing the limitations of several recently published approaches. For example, Cao et al. [1] and Wang et al. [2] enhanced detection accuracy but introduced heavier models, making them unsuitable for real-time edge deployment. On the other hand, He et al. [3] attempted to reduce complexity but at the cost of performance robustness. These comparisons allow us to better frame the challenge: existing fire detection algorithms either sacrifice accuracy for speed or vice versa, and few achieve both on embedded systems. To address this, we propose FCMI-YOLO, a real-time algorithm that strikes a better balance between detecti

---

## [Decision Letter · Decision Letter 1]

21 May 2025

PONE-D-25-01361R1FCMI-YOLO: A Efficient Deep Learning-Based Algorithm for Real-Time Fire Detection on Edge DevicesPLOS ONE

Dear Dr. Guan,

Thank you for submitting your manuscript to PLOS ONE. After careful consideration, we feel that it has merit but does not fully meet PLOS ONE’s publication criteria as it currently stands. Therefore, we invite you to submit a revised version of the manuscript that addresses the points raised during the review process.

**ACADEMIC EDITOR: **The minor comments made by the first author should be addressed in the best possible way.1- The authors should emphasize the merits of the developed method.

2-in the section simulation results, the proposed approach should compare with other relevant algorithms, the authors are asked to further provide comparisons to better show the advantages of their approach.

3. the literature review is not good enough.

4.some minor grammatical errors should be eliminated .the authors are kindly asked to double check the manuscript.

5.The abstract could be more concise and the point.

6.I suggest the authors consider exposes the equation more clearly so to achieve a broader audience.

7. please add more sentences to describe the limitations of this work. The advantages and disadvantages should be discussed.

We look forward to receiving your revised manuscript.

Kind regards,

Irving A. Cruz-Albarran

Academic Editor

PLOS ONE

Journal Requirements:

Additional Editor Comments :

Please address the comments raised by the reviewer. Please respond as clearly and concisely as possible.

Reviewers' comments:

Reviewer's Responses to Questions

**Comments to the Author**

1. If the authors have adequately addressed your comments raised in a previous round of review and you feel that this manuscript is now acceptable for publication, you may indicate that here to bypass the “Comments to the Author” section, enter your conflict of interest statement in the “Confidential to Editor” section, and submit your "Accept" recommendation.

Reviewer #1: All comments have been addressed

Reviewer #3: All comments have been addressed

2. Is the manuscript technically sound, and do the data support the conclusions?

Reviewer #1: Partly

Reviewer #3: Yes

3. Has the statistical analysis been performed appropriately and rigorously? 

Reviewer #1: Yes

Reviewer #3: Yes

4. Have the authors made all data underlying the findings in their manuscript fully available?

Reviewer #1: Yes

Reviewer #3: Yes

5. Is the manuscript presented in an intelligible fashion and written in standard English?

Reviewer #1: Yes

Reviewer #3: Yes

6. Review Comments to the Author

Reviewer #1: my recommendation is Minor (I THINK THIS CURRENT PAPER NOT ENOUGH FOR ACCEPTANCE).

1- The authors should emphasize the merits of the developed method.

2-in the section simulation results, the proposed approach should compare with other relevant algorithms, the authors are asked to further provide comparisons to better show the advantages of their approach.

3. the literature review is not good enough.

4.some minor grammatical errors should be eliminated .the authors are kindly asked to double check the manuscript.

5.The abstract could be more concise and the point.

6.I suggest the authors consider exposes the equation more clearly so to achieve a broader audience.

7. pleas add more sentences to describe the limitations of this work. The advantages and disadvantages should be discussed.

Reviewer #3: The main title is developed the idea very well. The main topic has been discussed by chronological order. The English is well written. The manuscript is following STEM very well. The manuscript has been developed well.

7. PLOS authors have the option to publish the peer review history of their article (what does this mean?). If published, this will include your full peer review and any attached files.

Reviewer #1: No

Reviewer #3: **Yes: **Norshida Abdul Kadir

---

## [Author Response · Author response to Decision Letter 2]

17 Jun 2025

We would like to thank the editor for handling the review process of our manuscript. We are very grateful to the reviewers for their thorough review and constructive comments. We have carefully addressed all the review comments and improved the quality of this paper accordingly. Please find below our response to review comments one by one.

In the Revised Manuscript with Track Changes, all modifications are highlighted in yellow for reference.

Response to Reviewer 1’s Comments

Comment 1: The authors should emphasize the merits of the developed method.

Response: We thank the reviewer for this valuable suggestion. In the revised manuscript, we have further emphasized the advantages of the proposed FCMI-YOLO algorithm from multiple aspects:

In the last paragraph of the “Related Work” section, we explicitly highlight the unique positioning of FCMI-YOLO compared to existing accuracy and lightweight algorithms, stating that it effectively achieves a trade-off between detection precision and inference speed, which is essential for edge deployment.

In Section 4.3.7 (Comparison of Mainstream Algorithms), we provide detailed comparisons with 17 mainstream algorithms in terms of precision, recall, inference speed, parameters, and FLOPs. These comparisons demonstrate that FCMI-YOLO outperforms most algorithms with only 4.2M parameters and 11.3G FLOPs, achieving 88.0% mAP@0.5 and 91.2 FPS. Additionally, we present visual detection comparisons under challenging conditions (e.g., occlusion, long-range), where our method shows superior robustness and minimal missed detections.

In Section 4.4.3 (Deployment Results), we detail outdoor experiments on the OrangePi 5 Plus, where FCMI-YOLO achieves higher detection accuracy than YOLOv5s at both 30m and 75m distances while maintaining real-time performance (23.4 FPS). These results clearly validate the algorithm’s practical effectiveness and suitability for deployment on resource-constrained platforms.

To further underscore these merits, we have added a summarizing paragraph at the end of Section 4.3.7 that articulates FCMI-YOLO’s balance between lightweight design, high accuracy, and real-time inference, especially under edge computing constraints. This helps to clearly position the proposed method in contrast to other mainstream algorithms.

Comment 2: In the section simulation results, the proposed approach should be compare with other relevant algorithms, the authors are asked to further provide comparisons to better show the advantages of their approach.

Response: We appreciate the reviewer’s insightful suggestion regarding the need to further highlight the advantages of the proposed algorithm through comparative analysis.

In response, we have significantly enhanced the simulation results in two key parts to address this issue:

1. Section 4.3.7 (Comparison of Mainstream Algorithms):

We conducted a detailed comparison between FCMI-YOLO and 17 mainstream algorithms, including Faster R-CNN, SSD, YOLOv3 to v11s variants, and several lightweight or enhanced versions such as YOLOv8s-World and YOLOv8s-FasterNet. Metrics such as model parameters, FLOPs, mAP@0.5, recall, and FPS were evaluated. The results show that FCMI-YOLO achieves the highest recall (84.4%) and maintains competitive precision (88.0%) while using significantly fewer parameters (4.2M) and computational cost (11.3 GFLOPs). As shown in Figure 13 of the revised manuscript, FCMI-YOLO exhibits more robust detection than other advanced YOLO variants under challenging fire scenarios.

2. Section 4.4.3 (Deployment Results):

To validate the algorithm’s effectiveness on edge devices, we performed deployment experiments on the OrangePi 5 Plus. We compared FCMI-YOLO against YOLOv5s at distances of 30 m and 75 m from fire sources. FCMI-YOLO achieved higher detection precision (88% and 70%) than YOLOv5s (72% and 59%), with only a minor decrease in FPS. These findings demonstrate the superior robustness and stability of our algorithm under resource-constrained devices.

Through these enhanced comparative experiments and visual analyses, we believe the advantages of FCMI-YOLO in terms of detection accuracy, real-time performance, and hardware adaptability are now more clearly demonstrated in the revised manuscript.

Comment 3: The literature review is not good enough.

Response: Thank you for your constructive feedback. In response to your suggestion, we have thoroughly revised the Related Work section to enhance its clarity, structure, and completeness.

The revised section now provides a more comprehensive overview of existing fire detection methods, beginning with a comparison between traditional handcrafted feature-based methods and deep learning-based methods. We further classify CNN-based detection algorithms into two-stage and one-stage algorithms and discuss their respective advantages and limitations in real-time fire detection applications.

In addition, we have reorganized and expanded the review of recent YOLO-based fire detection models. These works are now grouped according to their core contributions, including accuracy-oriented enhancements (for example, DG-YOLO, YOLOX-CBAM, FSDF, and YOLOv5 combined with EfficientDet) and lightweight optimization for edge deployment (such as Light-YOLOv4 and YOLO-ULNet).

Finally, we summarize the common trade-offs observed in current research, particularly the difficulty of balancing detection precision and computational efficiency, which motivates the development of the proposed FCMI-YOLO algorithm aimed at addressing these challenges on edge devices.

Comment 4: Some minor grammatical errors should be eliminated. The authors are kindly asked to double check the manuscript.

Response: We sincerely thank the reviewer for pointing this out. In response, we have carefully rechecked and polished the manuscript to improve its grammatical accuracy and language clarity. Special attention has been paid to sentence structure, verb usage, punctuation, and overall readability to enhance the quality and professionalism of the writing.

Comment 5: The abstract could be more concise and the point.

Response: Thank you for your valuable suggestion. In response, we have revised the abstract to make it more concise and focused on the core contributions of our work. Redundant descriptions have been removed, and the presentation of technical innovations and experimental results has been streamlined to enhance clarity and readability. The updated abstract clearly highlights the key components of our proposed method, namely FasterNext, CCFM, MLCA, and Inner-DIoU, and presents the performance improvements in a direct and succinct manner.

Original Abstract:

The rapid development of Internet of Things (IoT) technology and deep learning has propelled the deployment of vision-based fire detection algorithms on edge devices, significantly exacerbating the trade-off between accuracy and inference speed under hardware resource constraints. To address this issue, this paper proposes a real-time fire detection algorithm named FCMI-YOLO. Firstly, the FasterNext model, which is based on FasterNet, is introduced to reduce the computational of model parameters and enhance detection precision through lightweight design. Secondly, the Cross-Scale Feature Fusion Module (CCFM) and the Mixed Local Channel Attention (MLCA) mechanism are incorporated into the neck network, significantly improving detection performance for small fire targets and minimizing resource consumption. Finally, the Inner-DIoU loss function is proposed to optimize bounding box regression by incorporating auxiliary boxes. Experimental results on the fire dataset we constructed demonstrate that FCMI-YOLO increases mAP@50 by 1.5%, decreases the number of parameters by 40%, and lowers GFLOPs to 28.9% of those in YOLOv5s. These results demonstrate that FCMI-YOLO enhances fire detection accuracy while significantly reducing computational demands, making it highly valuable for widespread practical applications, particularly on resource-constrained edge devices. The core code and dataset required for the experiment are saved in this article at https://github.com/JunJieLu20230823/code.git.

Revised Abstract:

The rapid development of Internet of Things (IoT) technology and deep learning has propelled the deployment of vision-based fire detection algorithms on edge devices, significantly exacerbating the trade-off between accuracy and inference speed under hardware resource constraints. To address this issue, this paper proposes FCMI-YOLO, a real-time fire detection algorithm optimized for edge devices. Firstly, the FasterNext module is proposed to reduce computational cost and enhance detection precision through lightweight design. Secondly, the Cross-Scale Feature Fusion Module (CCFM) and the Mixed Local Channel Attention (MLCA) mechanism are incorporated into the neck network to improve detection performance for small fire targets and reduce resource consumption. Finally, the Inner-DIoU loss function is proposed to optimize bounding box regression. Experimental results on a custom fire dataset demonstrate that FCMI-YOLO increases mAP@50 by 1.5%, reduces parameters by 40%, and lowers GFLOPs to 28.9% of YOLOv5s, demonstrating its practical value for real-time fire detection in edge scenarios with limited computational resources. The core code and dataset are available at https://github.com/JunJieLu20230823/code.git.

Comment 6: I suggest the authors consider exposes the equation more clearly so to achieve a broader audience.

Response: Thank you very much for your insightful suggestion. In response, we have thoroughly revised the mathematical formulations and their corresponding explanations in the manuscript to enhance clarity and accessibility for a broader audience. Specifically, in Section 3.2 (FasterNext Network), we improved the presentation of the ReLU and SiLU activation functions by providing clearer definitions, highlighting their respective limitations, and explaining the rationale for adopting SiLU over ReLU. To aid intuitive understanding, a comparative plot is provided in Figure 4 of the revised manuscript and the accompanying text has been refined to more effectively capture the interest of readers. Additionally, in Section 3.4 (Improved Loss Function), we restructured the explanation of the CIoU, DIoU, and proposed Inner-DIoU loss functions. We clarified the roles and interactions of key parameters and emphasized their importance in fire detection tasks involving small or irregular targets. Moreover, we explicitly discussed the limitations of CIoU in such scenarios and the motivation for proposing Inner-DIoU. To further support understanding, Figure 7 in the revised manuscript presents a schematic diagram illustrating the concept of the inner bounding box used in the Inner-IoU calculation. These revisions aim to make the mathematical components of the manuscript more transparent and accessible to a wider range of readers.

Comment 7: Please add more sentences to describe the limitations of this work. The advantages and disadvantages should be discussed.

Response: Thank you for your valuable feedback. In response to your suggestion, we have revised the conclusion section to explicitly discuss the limitations of the proposed method and to present a balanced view of its advantages and disadvantages. Specifically, we have added three key limitations. First, in challenging environments such as strong wind or reflective surfaces, fire deformation, and mirror reflection of fire may interfere with feature extraction, leading to increased false positives and reduced detection accuracy. Second, although the proposed model has been optimized for lightweight deployment and achieves real-time performance on edge devices, its inference speed still requires further improvement to support more demanding real-time applications. Third, the performance of FCMI-YOLO may vary significantly across different hardware platforms, presenting a challenge for cross-platform optimization and deployment. In addition, we have clarified that future research will focus on enhancing robustness under complex environmental conditions, exploring more aggressive lightweight strategies to improve generalization and scalability.

---

## [Decision Letter · Decision Letter 2]

18 Jul 2025

FCMI-YOLO: An Efficient Deep Learning-Based Algorithm for Real-Time Fire Detection on Edge Devices

PONE-D-25-01361R2

Dear Dr. Guan,

We’re pleased to inform you that your manuscript has been judged scientifically suitable for publication and will be formally accepted for publication once it meets all outstanding technical requirements.

Kind regards,

Irving A. Cruz-Albarran

Academic Editor

PLOS ONE

Additional Editor Comments (optional):

Thank you very much for responding to the reviewers' comments appropriately.

Reviewers' comments:

Reviewer's Responses to Questions

**Comments to the Author**

1. If the authors have adequately addressed your comments raised in a previous round of review and you feel that this manuscript is now acceptable for publication, you may indicate that here to bypass the “Comments to the Author” section, enter your conflict of interest statement in the “Confidential to Editor” section, and submit your "Accept" recommendation.

Reviewer #1: All comments have been addressed

2. Is the manuscript technically sound, and do the data support the conclusions?

Reviewer #1: Yes

3. Has the statistical analysis been performed appropriately and rigorously? 

Reviewer #1: Yes

4. Have the authors made all data underlying the findings in their manuscript fully available?

Reviewer #1: Yes

5. Is the manuscript presented in an intelligible fashion and written in standard English?

Reviewer #1: Yes

6. Review Comments to the Author

Reviewer #1: After reviewing the revised version that was sent to them and considering the changes made, I believe the submission is acceptable from my point of view.

7. PLOS authors have the option to publish the peer review history of their article (what does this mean?). If published, this will include your full peer review and any attached files.

Reviewer #1: No

---

## [Editor Report · Acceptance letter]

PONE-D-25-01361R2

PLOS ONE

Dear Dr. Guan,

I'm pleased to inform you that your manuscript has been deemed suitable for publication in PLOS ONE. Congratulations! Your manuscript is now being handed over to our production team.

Kind regards,

on behalf of

Dr. Irving A. Cruz-Albarran

Academic Editor

PLOS ONE